# Discovering Symmetry Groups with Flow Matching

**Yuxuan Chen** [* 1]  **Jung Yeon Park** [† * 2]  **Floor Eijkelboom** [3]  **Jianke Yang** [4]  **Jan-Willem van de Meent** [3]
**Lawson L.S. Wong** [1]  **Robin Walters** [1]

## Abstract

Symmetry is fundamental to understanding physical systems and can improve performance and sample efficiency in machine learning. Both pursuits require knowledge of the underlying symmetries in data, yet discovering these symmetries automatically is challenging. We propose `LieFlow`, a novel framework that reframes symmetry discovery as a distribution learning problem on Lie groups. Instead of searching for the symmetry generators, our approach operates directly in group space, modeling a symmetry distribution over a large hypothesis group $G$. The support of the learned distribution reveals the underlying symmetry group $H \subseteq G$. Unlike previous works, `LieFlow` can discover both continuous and discrete symmetries within a unified framework, without assuming a fixed Lie algebra basis or a specific distribution over the group elements. Experiments on synthetic 2D and 3D point clouds, ModelNet10, and a real-world MI-Motion dataset show that `LieFlow` accurately discovers continuous and discrete subgroups, significantly outperforming a state-of-the-art baseline, LieGAN, in identifying discrete symmetries.

## 1 Introduction

Symmetry has long been central to mathematics and physics and, more recently, has featured prominently within machine learning. Classically, to model physical systems, the underlying symmetry is first identified and categorized; Noether's theorem (Noether, 1918) then establishes the existence of corresponding conservation laws, which can be used to understand and predict the system. In machine learn-

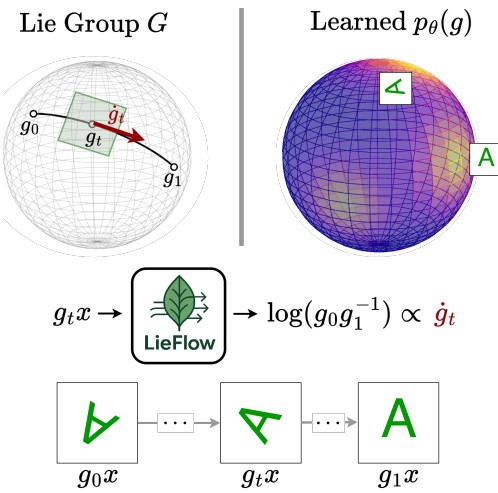

**Lie Group $G$**   **Learned $p_\theta(g)$**

$g_t x \rightarrow$ [LieFlow] $\rightarrow \log(g_0 g_1^{-1}) \propto \dot{g}_t$

$g_0 x$   $g_t x$   $g_1 x$

*Figure 1.* `LieFlow` discovers symmetries from data by assuming a hypothesis Lie group $G$ and learning the distribution of transformations over $G$ via flow matching over the Lie group, conditioned on data samples $x$.

ing, symmetries serve as powerful inductive biases in equivariant neural networks (Cohen & Welling, 2016; Kondor & Trivedi, 2018; Weiler & Cesa, 2019; He et al., 2021). These architectures leverage known symmetries in the data, such as rotational invariance in molecular structures (Thomas et al., 2018; Satorras et al., 2021) or translational equivariance in images (LeCun et al., 1998), to achieve superior performance with fewer parameters and training samples (Bronstein et al., 2021).

A critical limitation in both physics and machine learning is that the exact symmetry group must be known a priori. In practice, the underlying symmetries are often unknown, approximate, or domain-specific, e.g. hidden symmetries in physical systems (Gross, 1996; Liu & Tegmark, 2022), non-obvious chemical properties (Davies et al., 2016), and partial symmetries in 3D geometry (Mitra et al., 2006; 2013). Many applications can thus benefit from the ability to automatically discover and characterize symmetries from data, without relying on domain expertise or manual specification.

Several prior works have focused on symmetry discovery in restricted settings, such as roto-translations in images (Rao & Ruderman, 1998; Miao & Rao, 2007), commutative Lie groups (Cohen & Welling, 2014), or finite groups (Zhou

† Work done while at Northeastern University. [1]Northeastern University [2]Profluent Bio, Inc. [3]University of Amsterdam [4]University of California San Diego. Correspondence to: Jung Yeon Park <park.jungy@northeastern.edu>, Robin Walters <r.walters@northeastern.edu>.

*Proceedings of the 43$^{rd}$ International Conference on Machine Learning*, Seoul, South Korea. PMLR 306, 2026. Copyright 2026 by the author(s).

et al., 2020; Karjol et al., 2024). Recent studies on learning Lie groups also have some limitations: Benton et al. (2020) assume a fixed Lie algebra basis and a uniform distribution over the coefficients; Dehmamy et al. (2021) produce non-interpretable symmetries; Yang et al. (2023) assume a Gaussian distribution over the Lie algebra coefficients which is largely limited to continuous groups and use potentially unstable adversarial training; and Allingham et al. (2024) use a fixed Lie algebra basis and evaluate only over images.

In this work, we propose `LieFlow`, a new framework that fundamentally reframes symmetry discovery as a distribution learning problem on Lie groups. Conceptually, this shifts the paradigm of estimating generators to recovering group structure via support learning in group space. Specifically, we learn a distribution over a larger symmetry group, which serves as a hypothesis space of symmetries. The support of this distribution corresponds to the actual transformations observed in the data. We achieve this via flow matching directly on Lie groups (Figure 1).

Unlike standard flow matching (Lipman et al., 2023), which operates in data space, `LieFlow` learns a flow on a Lie group $G$ which is acting on the data space, conditioned on a data point, enabling the generation of new plausible data that respect the underlying symmetry structure. Thanks to the flexibility of flow matching, `LieFlow` can capture both continuous and discrete symmetries within a unified framework and accurately model the multi-modal nature of distributions supported on subgroups. We observe that discrete symmetry groups can be particularly challenging since cancellation among the different flow vectors results in small velocity until late in the flow, creating a "last-minute convergence" phenomenon. We counter this issue by proposing a novel time schedule for flow matching.

Qualitative and quantitative experimental results on synthetic 2D and 3D datasets, ModelNet10 (Wu et al., 2015) and a real-world MI-Motion dataset (Peng et al., 2023) show that `LieFlow` can discover discrete and continuous symmetries accurately, and achieve significant improvements over LieGAN (Yang et al., 2023), SGM (Allingham et al., 2024), and Augerino (Benton et al., 2020) baselines in identifying discrete symmetries.

Our key contributions are as follows:

- We propose a novel formulation of symmetry discovery as a flow matching problem on Lie groups, enabling the discovery of continuous, discrete, and partially observed symmetries.
- Extensive experiments on synthetic and real-world datasets demonstrate that `LieFlow` can accurately discover underlying symmetry groups, achieve significant improvements over LieGAN (Yang et al., 2023) baseline in identifying discrete symmetries, and are

- robust under noisy approximate symmetry settings.
- We identify a "last-minute convergence" phenomenon for discrete symmetries and propose a novel time scheduling method to address it.

## 2 Related Work

**Symmetry Discovery.** Many works on symmetry discovery differ on the types of symmetries that they can learn. Early works (Rao & Ruderman, 1998; Miao & Rao, 2007) used sequences of transformed images to learn Lie groups in an unsupervised way, but were limited to 2D roto-translations. Some recent works (Zhou et al., 2020; Karjol et al., 2024) can only discover finite groups, while others learn partially observed symmetries, i.e. subsets of known groups (Benton et al., 2020; Romero & Lohit, 2022; Chatzipantazis & Pertigkiozoglou, 2023). In particular, Augerino (Benton et al., 2020) requires labels to learn the invariant distribution of augmentations and is thus supervised, unlike our work. Shaw et al. (2024) discover continuous and discrete symmetries of machine learning functions beyond affine transformations, but focus on Killing type generators rather than probability flows on a Lie group. In contrast, `LieFlow` is flexible enough to learn both continuous and discrete symmetries, highly multi-modal distributions, and also partially observed symmetries via flow matching.

Several related works focus on continuous Lie groups but suffer from other limitations. Dehmamy et al. (2021) use Lie algebras in CNNs but learn the symmetries end-to-end along with the task function, leading to non-interpretable symmetries. Moskalev et al. (2022) extract Lie group generators from a neural network to analyze the learned equivariance, rather than discovering which symmetries exist in the dataset. Otto et al. (2025) discover symmetries of learned models via linear algebraic operators. Ko et al. (2024) learn infinitesimal generators of continuous symmetries from data via Neural ODEs and a task-specific validity score, capturing non-affine and approximate symmetries. Unlike these approaches, `LieFlow` models a full distribution over transformations within a prescribed matrix Lie group.

Yang et al. (2023) and Allingham et al. (2024) are most related to our work as they also learn distributions over transformations. Yang et al. (2023) propose LieGAN, which uses adversarial training to learn the Lie algebra basis while assuming a fixed distribution over the coefficients, typically Gaussian. This makes LieGAN well-suited for continuous symmetries; while it can discover discrete groups, doing so requires a carefully chosen prior distribution such as a multi-modal delta mixture. Allingham et al. (2024) first learn a canonicalization function to obtain a prototype as the orbit representative, then learn a generative model over transformations conditioned on the orbit representative. `LieFlow`, on the other hand, learns a marginal transformation distribution across all orbits directly via flow matching, without

assuming a fixed distribution or learning explicit prototypes.

**Generative Models on Manifolds.** Several recent works have extended continuous normalizing flows to manifolds (Gemici et al., 2016; Mathieu & Nickel, 2020; Falorsi, 2021) but rely on computationally expensive likelihood-based training. Some works (Rozen et al., 2021; Ben-Hamu et al., 2022; Chen & Lipman, 2024) have proposed simulation-free training methods; in particular, Chen & Lipman (2024) considers flow matching on general Riemannian manifolds. However, none of these works specifically consider Lie groups or the task of discovering symmetries from data. Other flow matching works directly incorporate known symmetries (Klein et al., 2023; Song et al., 2023; Bose et al., 2024) and do not infer symmetry from data. Another class of methods considers score matching and flow matching directly on Lie groups as data spaces (Zhu et al., 2025; Bertolini et al., 2026; Sherry & Smets, 2025). While methodologically similar to our work, they assume prior knowledge of the symmetry group and use specialized implementations for different Lie groups.

## 3 Preliminaries

**Lie Group.** A Lie group is a group that is also a smooth manifold, such that group multiplication and inversion are smooth maps. For example, the group of planar rotations $SO(2)$ is a Lie group parameterized by angle $\theta$, with elements $R_\theta = \left[ \begin{smallmatrix} \cos\theta & -\sin\theta \\ \sin\theta & \cos\theta \end{smallmatrix} \right]$. Each Lie group $G$ has an associated Lie algebra $\mathfrak{g}$, which is the tangent space at the identity. The Lie algebra captures the local (infinitesimal) group structure.

The exponential map $\exp\colon \mathfrak{g} \to G$ maps the Lie algebra to the Lie group. For matrix Lie groups, which we consider exclusively in this work, the exponential map is the matrix exponential $\exp(A) = \sum_{k=0}^{\infty} \frac{A^k}{k!}$. If the exponential map is surjective, we can restrict its domain and define the logarithm map $\log\colon G \supset U \to \mathfrak{g}$ for the neighborhood $U$ around the identity such that $\exp(\log(g)) = g$ for $g \in U$. Note that any Lie algebra element $A$ can be written as a linear combination of the Lie algebra basis. This allows us to generate the entire connected component of the identity in the Lie group using the Lie algebra, and thus the Lie algebra basis $L_i \in \mathfrak{g}$ is also called the (infinitesimal) generators of the Lie group.

**Flow Matching.** Flow Matching (FM) (Lipman et al., 2023; Liu et al., 2023; Albergo et al., 2025) is a scalable method for training continuous normalizing flows. The goal is to transport samples $x_0 \sim p_0$ drawn from a prior (e.g., Gaussian noise) to data samples $x_1 \sim p_1$. This transport is described by a time-dependent flow map[1] $\psi_t\colon \mathbb{R}^D \to \mathbb{R}^D$

---

[1]$\psi_t$ is technically an evolution operator arising from a time-dependent vector field; we follow standard terminology in the flow matching literature and use the term flow.

for $t \in [0, 1]$, where $\psi_t = \psi(t, x)$ and $\psi_0(x_0) = x_0$ and $\psi_1(x_0) \approx x_1$. This map is defined through the ODE $\frac{\mathrm{d}}{\mathrm{d}t}\psi_t(x) = u_t(\psi_t(x))$, where $u_t$ is a time-dependent velocity field governing the dynamics. To construct a training signal, FM interpolates between $x_0$ and $x_1$ over time, most commonly by way of a straight-line interpolation

$$x_t = (1-t)x_0 + tx_1, \quad \dot{x}_t = x_1 - x_0, \qquad (1)$$

defining a stochastic process $\{x_t\}$, also referred to as a *probability path*.

In Flow Matching, we define the *velocity field* $u_t$ that generates the desired probability path, and train a neural velocity field $v_t^\theta$ to match it:

$$\mathcal{L}_{\mathrm{FM}}(\theta) = \mathbb{E}_{t,x} \left\| v_t^\theta(x) - u_t(x) \right\|^2. \qquad (2)$$

Since $u_t$ can be written as the conditional expectation of these per-sample derivatives, $u_t(x) = \mathbb{E}[\dot{x}_t \mid x_t = x]$, we obtain the practical *Conditional Flow Matching* (CFM) loss

$$\mathcal{L}_{\mathrm{CFM}}(\theta) = \mathbb{E}_{t,x_0,x_1} \left\| v_t^\theta(x_t) - \dot{x}_t \right\|^2, \qquad (3)$$

where $x_t$ and $\dot{x}_t$ come from the chosen interpolation.

A key advantage of FM is that the transport dynamics can be learned entirely in a *self-supervised* manner, since interpolations between noise and data provide training targets without requiring access to likelihoods or score functions.

## 4 Symmetry Discovery via Flow Matching

### 4.1 Motivation and Formalism

A central challenge in symmetry discovery is not merely to estimate transformation parameters, but to determine which transformations are symmetries at all. In realistic settings, we are not given paired observations of original and transformed data, and the space of candidate transformations is typically much larger than the true symmetry group expressed in the data. We address this challenge by reframing symmetry discovery as a distribution learning problem on groups. Rather than directly estimating generators or invariants, we learn a probability distribution over a large hypothesis group $G$, whose support concentrates on the true symmetry subgroup $H \subseteq G$. We realize this idea using flow matching on Lie groups, which provides a flexible and self-supervised mechanism for learning which transformations preserve the data distribution.

**Problem Formulation.** Our goal is to recover the unknown group of transformations $H$ that preserves the data distribution $q$ on data space $\mathcal{X}$. We first assume a large hypothesis group $G$ acting on $\mathcal{X}$ that contains hypothetical symmetries. We wish to find the subgroup $H \subseteq G$ such that $q(hx) = q(x)$ for all $x \in \mathcal{X}$ and $h \in H$.

**Discovering Symmetry as Distribution Learning.** We frame the problem of finding $H$ as learning a distribution over the hypothesis group $G$ that concentrates on the subgroup $H$. To achieve this, we train an FM model that maps a chosen prior distribution over $G$ to a distribution $p_\theta(g)$ supported on $H$. The structure of the discovered symmetries emerges from the concentration pattern of $p_\theta(g)$: continuous symmetries yield a distribution spreading across the manifold of valid transformations, while discrete subgroups exhibit sharp peaks at a finite set of group elements (e.g. the four-fold rotations of $C_4$). Thus, LieFlow effectively filters the large hypothesis group down to the subgroup $H \subseteq G$ consistent with the data.

**Hypothesis Group and Prior over Symmetries.** We assume that the hypothesis group $G$ is connected, ensuring a well-defined logarithm map for compact groups. For the prior distribution $p$ over the group $G$, we use a uniform prior for compact groups and bounded priors over the Lie algebra coefficients for non-compact groups. When the exponential map is not surjective, we define the prior implicitly by sampling from the Lie algebra $\mathfrak{g}$ and applying the exponential map to construct group elements, ensuring well-defined training targets. In experiments, we consider $G = \mathrm{SO}(2)$ (planar rotations), $\mathrm{GL}(2)$ (invertible matrices) for the 2D datasets and $G = \mathrm{SO}(3)$ (3D rotations) for the 3D datasets.

**Interpolation Paths Along Orbits.** We now describe how to construct interpolation paths that respect the group structure. Consider a data sample $x_1 \sim q$ and a transformation $g \sim p(G)$ drawn from the prior distribution. We apply the transformation to obtain $x_0 = gx_1$, which serves as the starting point of our flow. We wish to learn to flow from $x_0$ to $x_1$ using only transformations in $G$. Let $A = \log(g^{-1}) \in \mathfrak{g}$ be the Lie algebra element that generates the inverse transformation. We define the interpolation in the data space via the group action:

$$x_t = \exp(tA)x_0, \quad t \in [0, 1], \quad (4)$$

At $t = 0$, we have $x_0 = gx_1$, and at $t = 1$, we recover $x_1$ since $\exp(A) = g^{-1}$. Marginalizing over $g$, this defines a conditional $p_t(x_t \mid x_1)$ that stays on the group orbit of $x_1$.

**The Target Velocity Field.** The derivative $\frac{d}{dt}x_t \in T_{x_t}\mathcal{X}$ defines a vector field along the path $\{x_t\}_{t \in [0,1]}$. Unlike standard flow matching, where the network outputs a velocity in data space, we configure our network to output a Lie algebra element $A \in \mathfrak{g}$ directly. That is, we set the conditional target field to $u_t(x_t \mid x_1) = A$.

This choice has two advantages. First, the update rule $x_{t+\Delta_t} = \exp(\Delta_t A_t)x_t$ ensures that the trajectory stays on the group orbit of $x_1$. Second, we can accumulate the transformations to obtain group elements in $H$ (Line 9, Algorithm 2). Note that $A$ is constant along the path, so the

network learns to predict the Lie algebra element generating the transformation from $x_0$ to $x_1$, regardless of $t$. The velocity field $\frac{d}{dt}x_t$ is obtained from $A$ via the differential of the group action; see Appendix A for more details.

**Objective.** The CFM objective on Lie groups is similar to the Euclidean case and is given by

$$\mathcal{L}_{\mathrm{LieCFM}} = \mathbb{E}_{t,q,p_t(x_t|x_1)} \|v_t^\theta(x_t) - u_t(x_t \mid x_1)\|_{\mathcal{G}}^2, \quad (5)$$

where $x_1 \sim q$, $x_t$ is defined by Eq. (4), $p_t(x_t \mid x_1)$ is the distribution of points along the paths, and $\mathcal{G}$ is a Riemannian metric on the Lie group, which equips the manifold with an inner product on each tangent space, allowing us to measure distances along the manifold. We use a left-invariant metric as it is completely determined by the inner product on the Lie algebra $\mathfrak{g}$ and the push-forward of the left action is an isometry, making computations simpler. The specific form of this metric is determined by the structure of the Lie group under consideration.

Although transformations are sampled from a broad hypothesis group during training, the flow-matching objective implicitly filters out transformations that are not true symmetries of the data. When a sampled transformation preserves the data distribution, applying it to a data point produces another statistically valid sample, and the resulting interpolation paths induce consistent training signals across many data points. In contrast, transformations that do not preserve the data distribution produce inconsistent targets: different data points yield incompatible directions, which cancel out when the model averages training signals. As a result, only transformations that are globally consistent with the data admit a coherent vector field, and the learned symmetry distribution concentrates its probability mass on the true symmetry group.

### 4.2 Training and Generation with LieFlow

**Training.** Using the interpolation paths and the Lie group flow matching objective, we can now derive the training and sampling algorithms for our method. The training algorithm (Algorithm 1) directly implements the objective by sampling

---

**Algorithm 1** Training

1: **repeat**
2:   $x_1 \sim q$
3:   $g \sim p(G)$
4:   $t \in [0, 1)$
5:   $x_0 = gx_1$
6:   $A = \log(g^{-1})$
7:   $x_t = \exp(tA)x_0$
8:   Take gradient descent step on $\nabla_\theta \|v_\theta(x_t, t) - A\|^2$
9: **until** converged

---

$x_1$ from the data distribution, sampling a transformation $g \sim p(G)$, and constructing $x_0 = gx_1$. The target Lie algebra group element $A$ can be computed analytically using the sampled group element $g$. We sample a timestep $t \in [0, 1)$ and construct a point $x_t = \exp(tA)x_0$ on the curve. Given $x_t$ and $t$, the model predicts $A$. We later show a modified time sampling procedure that works well for discovering discrete subgroups in Section 5.4.

**Sampling and Generation.** During sampling (Algorithm 2), we integrate over the learned vector field to produce a new sample $x'_1$. Starting from $x_0 = gx_1$ for a random $g \sim p(G)$, we iteratively apply the predicted Lie algebra elements using Euler integration with the exponential map $x'_{t+\Delta t} = \exp(\Delta_t A_t)x'_t$. This ensures that the trajectory remains on the group orbit.

---

**Algorithm 2** Generating data

---

**Require:** $x_1 \sim q$, number of steps $T$
1: $g \sim p(G)$
2: $x_0 = gx_1$
3: $\Delta_t = 1/T$
4: $M = I$
5: $x'_0 = x_0$
6: **for** $t = \{0, \Delta_t, 2\Delta_t, \ldots, (T-1)\Delta_t\}$ **do**
7:     $A_t = v^\theta(x'_t, t)$
8:     $x'_{t+\Delta_t} = \exp(\Delta_t A_t)x'_t$
9:     $M = \exp(\Delta_t A_t)M$     ▷ *Accumulate transforms*
10: **end for**
11: **return** $x'_1, M$

---

**Algorithm 3** Generating Group Elements

---

1: $x_1 \sim q$
2: $g \sim p(G)$
3: $x_0 = gx_1$
4: $x'_1, M$ from Alg. 2 Lines 3-10
5: **return** $Mg$

---

We also accumulate the product of all applied transformations into $M$ such that at the end of sampling, $x'_1 = Mx'_0$. The transformation $M$ is used for generating new group elements (Algorithm 3).

Algorithm 3 shows how to generate group elements $h$ consistent with the target group $H$. Given a data sample $x_1$, we transform it by $g \sim p(G)$ to obtain $x_0 = gx_1$. We then run Alg. 2 partially (lines 3-10) on $x_0$, obtaining outputs $x'_1$ and $M$. The key observation is that the composed transform $Mg$ forms a group element $h \in H$ that maps $x_1$ to $x'_1$.

We also prove that, under reasonable assumptions, Algorithm 3 generates group elements supported in the target group $H$:

**Proposition 4.1.** *Under the assumptions of ideal convergence, globally generated support, and the condition that point stabilizers are global symmetries, the group elements sampled by Algorithm 3 are supported in $H$.*

Complete assumptions and proof are deferred to Appendix B.

### 4.3 Handling Partially Observed Symmetries

The main assumption of this paper is that there exists a group $H$ to which the data distribution is invariant. This requires that the data be spread uniformly along the orbits of an entire group. In practice, however, data distributions can exhibit partial symmetries, i.e. $q(hx) = q(x)$ holds for only some $x \in \mathcal{X}$ and $h \in H$. We discuss how our method can be generalized to handle this case.

Formally, we consider the scenario where there exists a maximal subset $H_x \subseteq H$ for each $x \in \mathcal{X}$ such that for all $h \in H_x$, $q(hx) = q(x)$. For example, consider a data distribution of three arrow shapes: $q(\leftarrow) = q(\uparrow) = q(\rightarrow) = 1/3$, and the group $H = \langle h \rangle$, where $h$ acts on data by clockwise $\pi/2$ rotation. Then we have $H_\leftarrow = \{e, h, h^2\}$, $H_\uparrow = \{e, h, h^3\}$, $H_\rightarrow = \{e, h^2, h^3\}$. The goal of symmetry discovery then becomes, in addition to finding a subgroup $H \subset G$, identifying the input-dependent "symmetry subset" $H_x \subseteq H$ for any $x$.

Note that our method does not learn the explicit algebraic structure of $H$, but instead learns a distribution over the hypothesis group $G$ and identifies $H$ by (the closure of) the *support* of the learned distribution density. Analogously, to discover the partial symmetries stated above, it suffices to learn a conditional distribution over $G$, $p_\theta(\cdot|x)$, whose support at each $x$ reveals the symmetry subset $H_x$. Algorithm 3 can already learn this conditional distribution by fixing $x_1$ and repeatedly running the procedure with different $g \sim p(G)$. This produces samples of $p_\theta(\cdot \mid x_1)$, whose support estimates $H_{x_1}$.

## 5 Experiments

To evaluate `LieFlow`, we consider a variety of datasets consisting of 2D and 3D point clouds with different underlying ground truth symmetry groups. We also show our method can succeed with different hypothesis groups, in settings with partially observed symmetries where only a subset of group transformations are observed in data, and in settings where observations are noisy.

### 5.1 Datasets

We evaluate `LieFlow` on both synthetic 2D and 3D point clouds, ModelNet10 (Wu et al., 2015) which contains 3D models of real-world objects, and MI-Motion (Peng et al.,

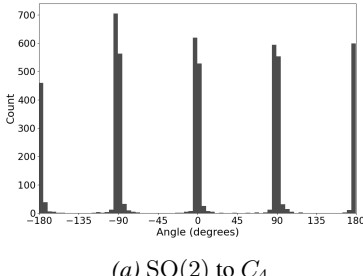

*(a)* SO(2) to $C_4$

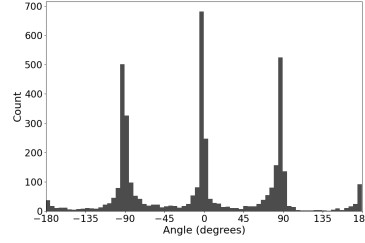

*(b)* $\mathrm{GL}(2, \mathbb{R})^+$ to $C_4$ (Lie algebra coefficients based sampling)

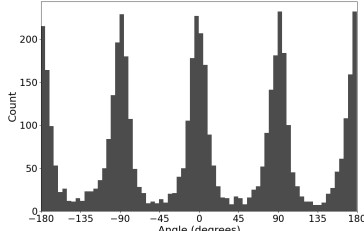

*(c)* $\mathrm{GL}(2, \mathbb{R})^+$ to $C_4$ (matrix composition based sampling)

*Figure 2.* Histograms of $5,000$ learned angles for 2D datasets, where the peaks correspond to the elements of the target symmetry group $C_4$. `LieFlow` successfully recovers $C_4$ symmetry group from (a) a compact hypothesis group SO(2) (b) a non-compact hypothesis group, (c) different sampling strategies.

2023) which contains motion capture data of human movements. For the synthetic point cloud and ModelNet10 experiments, we generate several datasets by transforming the canonical objects with group elements sampled from a target ground truth symmetry group $H$, see Figures 7, 8a. in Appendix C.

**Synthetic datasets** We consider two datasets, each generated from a single canonical object: a 2D arrow and a 3D irregular tetrahedron with no self-symmetries. For the 2D arrow, we add data points given by $H = C_4$ transformations and perform flow matching with hypothesis groups SO(2) and $\mathrm{GL}(2, \mathbb{R})^+$. For SO(2), we uniformly sample angles in $[0, 2\pi)$ to generate group elements. For $\mathrm{GL}(2, \mathbb{R})^+$, which is non-compact, we consider two different prior distributions to sample group elements : (1) **Lie algebra coefficients based sampling (L)**: uniformly sampling the coefficients of the four orthogonal basis vectors and applying the exponential map; (2) **Matrix composition based sampling (M)**: independently sampling rotation, scaling, and shear matrices and multiplying them. See Appendix D for more details. For the 3D dataset, we attempt to discover four different ground truth symmetry groups: tetrahedral Tet, octahedral Oct, icosahedral Ico and SO(2). We use SO(3) as a hypothesis group.

**ModelNet10** To demonstrate that `LieFlow` can handle complex real-world objects of diverse shapes, we consider ModelNet10 dataset(Wu et al., 2015). We first randomly select 10 objects from each of the 10 categories of ModelNet10 as canonical shapes, and downsample the point clouds to 128 points. See Figure 8a. in Appendix C for example samples. We consider five different target symmetry groups Tet, Oct, Ico, the SO(2) subgroup stabilizing the $z$ axis and SO(2) stabilizing $(0, \frac{1}{2}, -\frac{\sqrt{3}}{2})$.

**MI-Motion** To evaluate symmetry discovery in a real-world setting, we additionally consider the MI-Motion dataset (Peng et al., 2023), which contains sequences of 3D point clouds of human joints where multiple people inter-

act in diverse settings, such as indoor environments, parks, and complex crowds. Unlike the synthetic and ModelNet10 benchmarks, where the target group is used to generate transformed samples, we do not impose a target symmetry group on MI-Motion; it must be inferred from the data distribution itself. In this dataset, human trajectories are approximately canonicalized along the $x$ or $y$-axis in the world frame (people primarily move along axis-aligned directions), while gravity breaks full SO(3) symmetry. This induces an approximate $C_4$ rotational symmetry around the vertical $z$-axis, with deviations caused by interactions and scene variability. We randomly sample 500,000 3D skeletons from the dataset and apply `LieFlow` to discover the underlying symmetry structure. See Figure 8b in Appendix C for example samples. We use SO(3) as the hypothesis group.

## 5.2 Evaluation

In order to properly measure the quality of discovered symmetries, we propose using the Wasserstein-1 distance between the generated group elements (sampled via Algorithm 3) and the ground truth elements. Given two distributions $P$ and $Q$ over a metric space $(\mathcal{M}, d_G)$, let $\Pi(P, Q)$ be the set of all couplings between $P$ and $Q$. The Wasserstein-1 distance is defined as

$$W_1(P, Q) = \inf_{\gamma \in \Pi(P,Q)} \mathbb{E}_{(x,y) \sim \gamma}[d_G(x, y)], \quad (6)$$

and it can be computed using the `POT` Python package.

Note that $W_1$ values alone are not always comparable across hypothesis groups with different geometries. To address this issue, we additionally propose the relative improvement over random sampling,

$$\mathrm{RI} = 1 - \frac{W_1}{W_{\mathrm{random}}},$$

where $W_{\mathrm{random}}$ is computed from uniformly sampled elements of the hypothesis group.

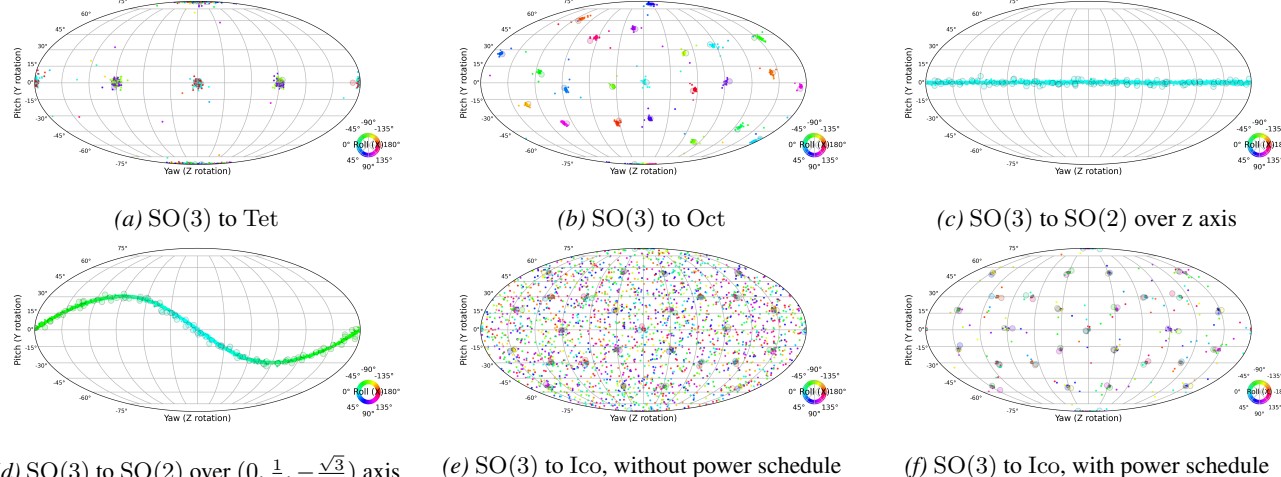

*(a)* SO(3) to Tet          *(b)* SO(3) to Oct          *(c)* SO(3) to SO(2) over z axis

*(d)* SO(3) to SO(2) over $(0, \frac{1}{2}, -\frac{\sqrt{3}}{2})$ axis          *(e)* SO(3) to Ico, without power schedule          *(f)* SO(3) to Ico, with power schedule

*Figure 3.* 3D Datasets: Visualization of 5,000 generated elements of SO(3) by Euler angles. The first two angles are represented spatially on the sphere using Mollweide projection and the color represents the third angle. The elements are canonicalized by the original random transformation and the ground truth elements of the target group are shown in circles with gray borders. The points are jittered with uniformly random noise to prevent overlapping. Our method recovers a variety of ground truth symmetry groups.

This metric normalizes for group-dependent scale: RI = 1 suggests perfect recovery, and negative values indicate the performance is worse than random.

In the 2D experiments, we use the Euclidean distance as the metric $d_G$ on the group. In the 3D experiments, we use the geodesic distance on SO(3), which is given by $d_G(R_1, R_2) = \arccos((\text{tr}(R_1^T R_2) - 1)/2)$, where $R_1$ and $R_2$ are two rotation matrices.

As a baseline, we compare against LieGAN (Yang et al., 2023), a recent state-of-the-art method for learning Lie group symmetries using a GAN. We set the number of generators in LieGAN to be the same as the dimension of our hypothesis group, and use a uniform distribution for the coefficients since it performs better than a Gaussian distribution in our experiments. We also keep the discriminator architecture the same as our predictive network for a fair comparison. On ModelNet10, we additionally compare against Augerino (Benton et al., 2020) and SGM (Allingham et al., 2024). Since Augerino requires labels, we assign the same label to all samples on the same canonical object orbit. The network architecture of Augerino and SGM is the same as the prediction network in `LieFlow`, and we use SO(3) as the hypothesis group for a fair comparison. We report results averaged over 3 random seeds and their standard deviations. See Appendix D for more training details.

### 5.3 Results on Synthetic 2D Datasets

Figures 2a, 2b, and 2c show the histograms of learned angles for the 2D dataset with target group $C_4$. All histograms correctly display four peaks at the $C_4$ group elements, confirming that `LieFlow` discovers the underlying symmetries across different hypothesis groups and prior distributions.

*Table 1.* Wasserstein-1 distance and relative improvement over random sampling between generated group elements and ground truth group elements in 2D experiments. We report $W_1$ (RI), where lower $W_1$ and higher RI are better. We use 1 and 4 LieGAN generators, corresponding to the dimension of SO(2) and GL$(2, \mathbb{R})^+$, respectively. Note that LieGAN has the same $W_1$ values across the row as it does not depend on hypothesis groups. (L) and (M) indicate Lie algebra coefficients based sampling and matrix composition based sampling, respectively.

| Method | SO(2) → $C_4$ | GL$(2,\mathbb{R})^+$ → $C_4$(L) | GL$(2,\mathbb{R})^+$ → $C_4$(M) |
|---|---|---|---|
| LieGAN (1.gen) | $1.714 \pm 0.031$ $(-216.1\%)$ | $1.714 \pm 0.031$ $(80.5\%)$ | $1.714 \pm 0.031$ $(-93.8\%)$ |
| LieGAN (4.gen) | $1.073 \pm 0.067$ $(-97.9\%)$ | $1.073 \pm 0.067$ $(87.8\%)$ | $1.073 \pm 0.067$ $(-21.3\%)$ |
| LieFlow(Ours) | $\mathbf{0.054 \pm 0.012}$ $\mathbf{(90.0\%)}$ | $\mathbf{0.926 \pm 0.087}$ $\mathbf{(89.5\%)}$ | $\mathbf{0.505 \pm 0.040}$ $\mathbf{(42.9\%)}$ |

Table 1 shows the Wasserstein-1 distance between the generated group elements and the ground truth. `LieFlow` outperforms LieGAN in all settings. We can see that using a smaller hypothesis group SO(2) leads to much better performance, matching our intuition that a more informative prior is beneficial. Even with the larger GL(2) hypothesis group, both sampling settings outperform LieGAN.

**Partially Observed Symmetries.** We first consider the setting where only a subset of the transformations of the target group $C_4$ are present in the dataset, i.e. $C_4$ is only partially observed. Specifically, we construct the dataset by considering only $0°, 90°, 270°$ rotations to the canonical arrow object with equal probability. The angle histograms of the input-dependent distribution are shown in Figure 20. For each type of arrow $x_1$, `LieFlow` successfully identifies the corresponding symmetry subset $H_{x_1}$, which contains three out of the four $C_4$ group elements.

We further evaluate a functional partial symmetry setting by learning symmetries of the joint distribution $p(x, y)$. Specifically, we assign label 0 to arrows at orientations $0°$ and $180°$, and label 1 to arrows at orientations $90°$ and $270°$.

*Table 2.* Wasserstein-1 distance and relative improvement over random sampling in rotated ModelNet10 (Section 5.5) experiments. We report $W_1$ (RI%), where lower $W_1$ and higher RI are better. SO(3) → SO(2)(a) corresponds to rotations around $z$ axis and SO(3) → SO(2)(b) corresponds to rotations around $(0, 1/2, -\sqrt{3}/2)$ axis.

| Method | SO(3) → Tet | SO(3) → Oct | SO(3) → SO(2)(a) | SO(3) → SO(2)(b) | SO(3) → Ico |
|---|---|---|---|---|---|
| LieGAN | $1.113 \pm 0.024$ ($-23.5\%$) | $1.145 \pm 0.042$ ($-59.8\%$) | $0.101 \pm 0.032$ ($93.6\%$) | $0.072 \pm 0.024$ ($95.4\%$) | $0.893 \pm 0.029$ ($-72.2\%$) |
| SGM | $0.915 \pm 0.012$ ($-1.5\%$) | $0.742 \pm 0.017$ ($-3.6\%$) | $1.848 \pm 0.051$ ($-17.2\%$) | $1.832 \pm 0.045$ ($-16.2\%$) | $1.031 \pm 0.096$ ($-98.8\%$) |
| Augerino | $1.059 \pm 0.086$ ($-17.5\%$) | $1.083 \pm 0.094$ ($-51.2\%$) | $1.236 \pm 0.040$ ($21.6\%$) | $1.244 \pm 0.035$ ($21.1\%$) | $0.795 \pm 0.066$ ($-53.3\%$) |
| LieFlow(Ours) | $\mathbf{0.149 \pm 0.013}$ ($\mathbf{83.5\%}$) | $\mathbf{0.097 \pm 0.018}$ ($\mathbf{86.5\%}$) | $\mathbf{0.050 \pm 0.009}$ ($\mathbf{96.8\%}$) | $\mathbf{0.053 \pm 0.007}$ ($\mathbf{96.6\%}$) | $\mathbf{0.129 \pm 0.005}$ ($\mathbf{75.1\%}$) |

We concatenate a one-hot label representation to the point coordinates, forming a 4D point cloud, and use SO(2) × SO(2) as the hypothesis group, where the first factor acts on spatial coordinates and the second on the label dimension. As shown in Figure 21, the learned conditional distribution separates label-preserving transformations from label-changing ones, recovering the functional partial symmetry.

### 5.4 Results on Synthetic 3D Datasets

Figure 3 visualize generated group elements for the different target symmetry groups on SO(3), where the points are generated from our flow model and the ground truth symmetries are shown as circles with gray borders. For the tetrahedral group (Figure 3a), octahedral group (Figure 3b), SO(2) group around the $z$-axis (Figure 3c), and SO(2) group around $(0, \frac{1}{2}, -\frac{\sqrt{3}}{2})$ (Figure 3d), we can see that our model learns to correctly output transformations very close to the target group elements, similar to the 2D case. We find that training with a uniform time sampling procedure in Algorithm 1 fails to recover the icosahedral group and produces almost uniformly distributed elements over SO(3) (Figure 3e). We provide an improved sampling method in the next paragraph.

Table 5 presents quantitative results, where `LieFlow` significantly outperforms LieGAN in all discrete group while achieving comparable performance on continuous groups.

**Improved Time Sampling.** To further analyze the failure of uniform time sampling, we first plot the time progression of transformed objects in Figure 4a and Figure 4b. For discrete groups, the model only begins moving objects towards the group elements when $t$ is close to 1, whereas for continuous groups, the transformation occurs gradually throughout the flow. We find that the entropy of the posterior remains stationary until $t$ approaches 1 (see Appendix E for more details). This phenomenon of "last-minute convergence" makes learning discrete groups particularly challenging as the model needs to learn to produce a near-zero vector field for most time steps and only moves towards the correct mode at the end of inference.

Since most of the learning signal for discrete groups occurs near $t = 1$, we leverage a time scheduling scheme to sample more timesteps near $t = 1$. Instead of sam-

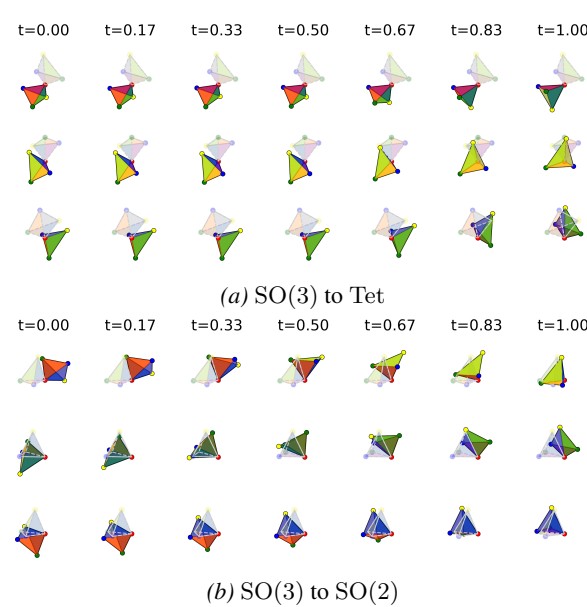

*(a)* SO(3) to Tet

*(b)* SO(3) to SO(2)

*Figure 4.* Time progression of $x_t$ when generating 3 samples over 6 steps to illustrate the inference process. The object starts moving at later time steps in the discrete group case ((a)), whereas it gradually moves at each time step in the continuous group case ((b)). The gray object shows the original $x_1$.

pling from $\mathcal{U}(0, 1)$, we use a power distribution with density $p(x) = nx^{n-1}, x \sim \mathcal{U}(0, 1)$, where $n$ controls the skewness (see Figure 15 in Appendix F). We find empirically that $n = 5$ works best; see Appendix F.1. Figure 16 in Appendix F confirms that with the power time schedule, objects begin moving earlier in the flow. With this power schedule, our model correctly learns the icosahedral group as shown in Figure 3f.

### 5.5 Results on ModelNet10

Table 2 shows the results for different target symmetry groups of real-world ModelNet10 objects, with qualitative visualizations in Appendix G.4. As with the synthetic 3D datasets, `LieFlow` significantly outperforms SGM and Augerino baselines, and LieGAN on all discrete groups while achieving comparable performance for continuous groups, demonstrating its effectiveness in discovering symmetries in real-world datasets. We again find that the power time schedule improves performance for the discrete groups.

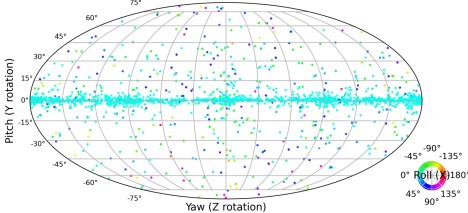

*Figure 5.* Distribution of generated SO(3) elements for MI-Motion dataset.

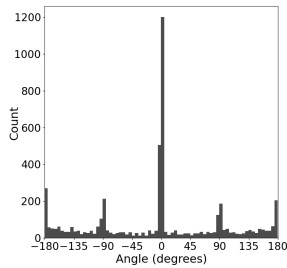 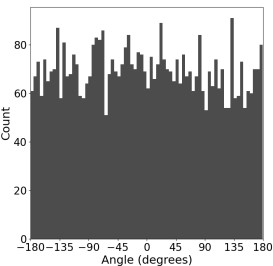

*(a)* LieFlow results      *(b)* LieGAN results

*Figure 6.* Histograms of the $z$-axis rotation angles extracted from generated SO(3) elements for MI-Motion dataset.

**Robustness Analysis.** In real-world datasets, noise may be present in the form of incomplete observations or small perturbations that break exact symmetries. To evaluate the robustness of `LieFlow` to noise, after the true symmetry transformation, we randomly mask a portion of the points and apply SO(3) perturbation $\exp(\hat{\omega})$ where $\omega \sim \mathcal{N}(0, \sigma^2 I)$ is a rotation vector and $\hat{\omega} \in \mathfrak{so}(3)$ is its skew-symmetric matrix. Table 3 shows the results for the challenging setting of discovering Ico with hypothesis group SO(3) with different masking ratios (0%, 5%, 10%) and different magnitudes of perturbation $\sigma$ (0, 0.05, 0.1). LieFlow degrades gracefully in noisy settings, with performance remaining reasonable even at about $\sigma = 0.1$ combined with 10% masking. See Figure 28 in the Appendix G.4 for visualizations.

*Table 3.* Robust analysis: Wasserstein-1 distance between generated group elements and ground truth group elements for SO(3) → Ico (lower is better). The columns correspond to different magnitudes of perturbation in $\sigma$ (0, 0.05, 0.1) while the rows correspond to different masking ratios (0%, 5%, 10%).

| Noises | 0% | 5% | 10% |
|---|---|---|---|
| 0 | $0.129 \pm 0.005$ | $0.133 \pm 0.008$ | $0.131 \pm 0.007$ |
| $0.05(\sim 3°)$ | $0.191 \pm 0.005$ | $0.202 \pm 0.006$ | $0.205 \pm 0.013$ |
| $0.1(\sim 6°)$ | $0.295 \pm 0.011$ | $0.306 \pm 0.006$ | $0.311 \pm 0.008$ |

### 5.6 Results on MI-Motion

Figure 5 visualizes the generated SO(3) elements for the MI-Motion dataset, where we can see that the learned transformation samples are mainly concentrated on rotations around the $z$-axis. We further extract the $z$-axis rotation component and plot the angle histogram in Figure 6. We see that `LieFlow` successfully captures the approximate $C_4$ symmetry around the $z$-axis, with peaks around $0, 90, 180, 270$ degrees, while LieGAN only learns a uniform distribution and fails to capture the correct structure.

## 6 Conclusion and Limitations

Our work demonstrates that expressive generative models such as flow matching provide a promising avenue for symmetry discovery. We introduce `LieFlow`, a framework that formulates symmetry discovery as learning a distribution over a hypothesis Lie group $G$, whose support reveals the underlying symmetry subgroup $H \subseteq G$. Unlike prior approaches that assume a fixed Lie algebra basis or a prescribed coefficient distribution, `LieFlow` learns directly in group space and can recover both continuous and discrete symmetries within a unified framework. Experiments on synthetic point clouds, ModelNet10 shapes, and real-world motion data show that `LieFlow` discovers a variety of symmetry structures, including $C_4$, partial $C_4$, finite subgroups of SO(3), and continuous SO(2) subgroups. The method also shows robustness under incomplete or noisy observations, and superior performance to several baselines.

While `LieFlow` advances the state of the art in symmetry discovery, it also has several limitations worth noting, which may provide opportunities for future work. First, `LieFlow` requires choosing a hypothesis group $G$. A tighter group provides a stronger inductive bias and can improve convergence, while overly broad or higher-dimensional groups may increase sampling and optimization difficulty. Scaling to large, high-dimensional, or non-compact hypothesis groups remains an important direction for future work. Second, our theory assumes that Lie algebra targets are identifiable, which holds when the infinitesimal stabilizer is trivial and when point stabilizers do not introduce ambiguities outside the global symmetry group. Continuous stabilizers may lead to non-unique Lie algebra coordinates and may require quotient formulations or auxiliary supervision. Third, our implementation relies on exponential and logarithm maps. While stable in our experiments on compact groups such as SO(2) and SO(3), logarithm branch ambiguities and numerical issues may arise near cut loci for more general groups. Finally, `LieFlow` learns a distribution over transformations rather than an explicit algebraic presentation of the discovered group. Extracting algebraic structure like infinitesimal generators from the learned distribution is a natural direction for future work, potentially using tools from computational group theory. Other important directions include incorporating weak supervision, extending to more data modalities, and evaluating the utility of the discovered symmetries in downstream equivariant models.

## Acknowledgements

The authors are grateful to Nima Dehmamy, Minghan Zhu, Xiaogang Peng and Rose Yu for helpful discussions. F.E. and J.W.M. would like to acknowledge support from the Bosch Center for Artificial Intelligence. J.W.M. acknowledges additional support from the European Union's Horizon Framework Programme (Grant agreement ID: 101120237). L.L.S.W. would like to acknowledge support from NSF Grants 2107256. R.W. would like to acknowledge support from NSF Grants 2442658 and 2134178. This work is supported by the National Science Foundation under Cooperative Agreement PHY-2019786 (The NSF AI Institute for Artificial Intelligence and Fundamental Interactions, http://iaifi.org/).

## Impact Statement

A potential risk of such automatic symmetry discovery is that the inferred symmetries may be incorrect or biased by hypothesis group $G$. If such symmetries are treated as exact invariances, they may propagate wrong inductive biases into downstream equivariant models or lead to misleading scientific interpretations. We therefore recommend validating discovered symmetries against domain knowledge before using them for deployment or scientific conclusions.

## LLM Usage

LLMs were used to revise sentences, correct grammar, and draft portions of this work. They were also used to generate parts of the model code and most of the visualization code.

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

# A  Derivation of the Target Flow Field

This section provides a detailed derivation of the target vector field $u_t(x_t \mid x_1)$ used in our Lie group flow matching framework. We begin by introducing flow matching on Lie groups, followed by the derivation of the target vector field on the data space induced by the group action.

**Interpolants on Lie Groups.**  Denote the left action of $G$ on itself for any $g \in G$ as $L_g : G \to G, \hat{g} \mapsto g\hat{g}$. Let $T_g G$ be the tangent space at $g \in G$. The push-forward of the left action is $(L_g)_* : T_{\hat{g}} G \to T_{g\hat{g}} G$, which transforms tangent vectors at $T_{\hat{g}} G$ to tangent vectors at $g\hat{g} \in G$ via the left action.

To derive FM on a Lie group, we need to define the interpolant as in the Euclidean case. An exponential curve starting at $g_0, g_1 \in G$ can be defined by (Sherry & Smets, 2025):

$$\gamma : [0, 1] \to G; \quad t \mapsto g_0 \exp(t \log(g_0^{-1} g_1)). \tag{7}$$

This curve starts at $\gamma(0) = g_0$ and ends at $\gamma(1) = g_1$. The multiplicative difference is $g_0^{-1} g_1$ and the logarithm maps it to the Lie algebra, giving the direction from $g_0$ to $g_1$. The scaled version is then exponentiated, recovering intermediate group elements along this curve. Using these exponential curves, we can define the target vector field as

$$u_t(g_t \mid g_1) = \frac{(L_{g_t})_* \log(g_t^{-1} g_1)}{1 - t}. \tag{8}$$

**Defining the Target Vector Field.**  Note that we couple source and target by an explicitly sampled group action. According to (4), given $x_1 \sim q$ and $g \sim p(G)$, and set $x_0 = g x_1$ and $A := \log(g^{-1}) \in \mathfrak{g}$. Consider the group curve $g_t = g \exp(tA)$ so that $g_t^{-1} = \exp((1-t)A)$ and the induced path $x_t = g_t x_1 = x_0 \exp(tA)$. Plugging $g_1 = e$ into the group interpolant (8), we have:

$$u_t(g_t \mid e) = \frac{(L_{g_t}) * \log(g_t^{-1})}{1 - t} = (L_{g_t}) * A, \tag{9}$$

which is a time independent target field in the Lie algebra along this curve. To obtain the conditional target on the data path, we first define the orbit map $\phi_{x_1} : G \to \mathcal{X}, \ \phi_{x_1}(g) = g x_1$. Its differential at $g_t$ is given as $(d\phi_{x_1})_{g_t} : T_{g_t} G \longrightarrow T_{x_t} \mathcal{X}$, where $x_t = \phi_{x_1}(g_t)$. Note that $(L_{g_t})* : \mathfrak{g} \to T_{g_t} G$. Compose them together, we obtain the pushforward operator $J_{x_t} := (d\phi_{x_1})_{g_t} \circ (L_{g_t})* : \mathfrak{g} \longrightarrow T_{x_t} \mathcal{X}$. Then, we push the group target to the data path and obtain the conditional target on data:

$$u_t(x_t \mid x_1) = (d\phi_{x_1})_{g_t} \big[ u_t(g_t \mid e) \big] = (d\phi_{x_1})_{g_t} \big[ (L_{g_t}) * A \big] = J_{x_t}(A) \ \in \ T_{x_t} \mathcal{X}. \tag{10}$$

We define the stabilizer of $x_t$ as $\mathrm{Stab}_G(x_t) = \{ s \in G : s \cdot x_t = x_t \}$, and denote its Lie algebra by $\mathfrak{stab}_{x_t} = \mathrm{Lie}(\mathrm{Stab}_G(x_t))$. The infinitesimal action map $J_{x_t} : \mathfrak{g} \to T_{x_t}(G \cdot x_1)$ satisfies $\ker J_{x_t} = \mathfrak{stab}_{x_t}, \mathrm{Im}\, J_{x_t} = T_{x_t}(G \cdot x_1)$. Thus, $J_{x_t}$ is always surjective onto the orbit tangent space, and it is injective precisely when the infinitesimal stabilizer is trivial, i.e., $\mathfrak{stab}_{x_t} = \{0\}$. Under this condition, each tangent vector along the orbit has a unique Lie algebra coordinate $A \in \mathfrak{g}$, so we can identify $u_t(x_t \mid x_1) = J_{x_t}(A)$ with its unique coordinate $A$ and use $A$ as the target in the Lie algebra regression loss.

Note that finite stabilizers introduce no Lie algebra degeneracy, since their Lie algebra is zero. Therefore, objects with discrete self-symmetries can still admit unique infinitesimal targets. Experiment on $C_4$ rectangles in Appendix G.1 confirms that `LieFlow` can successfully learn the correct $C_4$ symmetry even though each rectangle has discrete nontrivial $C_2$ stabilizer. Ambiguity arises only when the stabilizer contains a continuous subgroup, in which case $\mathfrak{stab}_{x_t} \neq \{0\}$ and the coordinate $A$ is identifiable only modulo $\mathfrak{stab}_{x_t}$. In such cases, the velocity $J_{x_t}(A)$ remains well-defined, but its Lie algebra coordinate is not unique without an additional gauge choice, quotient formulation, or auxiliary information.

# B   Support recovery by `LieFlow`

**Data distribution and global symmetry group.**   Let $G$ be a group acting on the data space $\mathcal{X}$, and let $q$ be a data density on $\mathcal{X}$ with support

$$S := \mathrm{supp}(q).$$

The action of $G$ on $\mathcal{X}$ induces an action on densities by pushforward. We define the global data-distribution symmetry group as

$$H := \mathrm{Stab}_G(q) = \{g \in G : gq = q\}.$$

Equivalently, $H$ consists of transformations in the hypothesis group $G$ that preserve the whole data distribution.

**Assumption B.1** (Ideal convergence and support consistency). Let $p_\theta(h \mid x)$ be the conditional distribution over group elements induced by Algorithm 3, and define the generated distribution

$$\hat{q}_\theta(x') = \int_\mathcal{X} \int_G \delta(x' - hx)\, p_\theta(h \mid x) q(x)\, d\mu(h) dx.$$

We assume the flow matching objective is realizable and optimized to its ideal optimum, so that $\hat{q}_\theta = q$. Consequently at the support level, generated samples remain in the data support:

$$hx \in S := \mathrm{supp}(q)$$

for $q(x)p_\theta(h \mid x)$-almost every pair $(x, h)$. Under the standard topological definition of support and continuity of the group action, we use the corresponding support statement: for $q$-almost every $x \in S$,

$$h \in \mathrm{supp}\, p_\theta(\cdot \mid x) \quad \Longrightarrow \quad hx \in S.$$

Assumption B.1 is the support-level consequence of ideal flow matching convergence. If the learned sampler induces exactly the data distribution, then it should not assign probability to transformations that send data points outside the observed data support. Otherwise, the generated distribution $\hat{q}_\theta$ would place mass outside $S = \mathrm{supp}(q)$ and could not equal $q$. This assumption states that, at the ideal optimum, sampled transformations are compatible to data.

**Assumption B.2** (Globally generated support). For every $x \in S$,

$$(Gx) \cap S = Hx.$$

That is, within the $G$-orbit of any observed point $x$, the points that remain in the data support are exactly those obtained by applying elements of the global symmetry group $H$.

This assumption formalizes the clean exact global-symmetry setting: the same symmetry group $H$ acts consistently on every observed $G$-orbit, and the observed support contains the full $H$-orbit rather than only a partial or approximately symmetric subset. That is, the symmetry is consistent across the entire data domain. It may fail when facing partial or approximate symmetries. For example, given $G$ as $\mathrm{SO}(2)$, if one orbit in the data exhibits $C_4$ symmetry while another orbit exhibits only $C_2$ symmetry, then the common global symmetry $H$ ($\mathrm{Stab}_G(q) = \{g \in G : gq = q\}$) should be their intersection $C_2$, whereas aggregating conditional samples across orbits may also include transformations that are valid only locally for the $C_4$ orbit. While making this assumption somewhat reduces our ability to detect local symmetry, it represents a reasonable trade off; by coarsening the discovery problem we magnify the signal. We use this assumption as an inductive bias for support recovery: when the data are generated by a single global symmetry group, aggregating conditional transformation samples across data points should reinforce the shared group structure rather than mix incompatible local structures.

**Assumption B.3** (Point stabilizers are global symmetries). For every $x \in S = \mathrm{supp}(q)$, the point stabilizer of $x$ in the hypothesis group $G$ is a subgroup of the global symmetry group $H$:

$$\mathrm{Stab}_G(x) \subseteq H,$$

where $\mathrm{Stab}_G(x) = \{g \in G : gx = x\}$. Equivalently, any transformation in the hypothesis group that leaves an observed sample fixed is already a global symmetry of the data distribution.

Assumption B.3 rules out point-wise ambiguities that are not true global symmetries. From Assumption B.2 and support consistency, we can conclude that a sampled transformation $h$ maps $x$ to the same orbit point as some global symmetry $h_0 \in H$, i.e., $hx = h_0 x$. However, this only implies that $h$ and $h_0$ differ by an element of the point stabilizer $\mathrm{Stab}_G(x)$. If $\mathrm{Stab}_G(x)$ contained transformations outside $H$, Algorithm 3 could assign support to transformations that fix a particular sample but do not preserve the full data distribution. For example, suppose the global symmetry group is $H = C_2$ under planar rotations, but one observed orbit contains an equilateral triangle. This triangle has a larger point stabilizer $C_3$: rotations by $120°$ and $240°$ leave this particular sample unchanged. These $C_3$ rotations are valid for that sample, since they map $x$ to itself, but they are not global symmetries of the whole dataset if other samples only respect $C_2$. In this case, aggregating conditional samples could introduce transformations outside $H$. This assumption excludes this failure mode by requiring all such point stabilizers to already be global symmetries.

**Theorem B.4** (Support recovery by `LieFlow`). *Suppose Assumptions B.1, B.2, and B.3 hold. Then, for q-almost every $x \in S$,*

$$\mathrm{supp}\, p_\theta(\cdot|x) \subseteq H.$$

*Consequently, the marginal group-element distribution sampled by Algorithm 3,*

$$p_\theta(\cdot) = \int_{\mathcal{X}} p_\theta(\cdot|x)q(x)dx,$$

*also satisfies*

$$\mathrm{supp}\, p_\theta \subseteq H.$$

*Proof.* Fix any $x \in S$ for which the support-consistency condition holds, and take any group element

$$h \in \mathrm{supp}\, p_\theta(\cdot|x).$$

By assumption B.1, applying $h$ to $x$ produces a point in the data support $hx \in S$. Since $h \in G$, the point $hx$ also lies in the $G$-orbit of $x$, that is $hx \in Gx$. Therefore,

$$hx \in (Gx) \cap S.$$

By assumption B.2,

$$(Gx) \cap S = Hx.$$

Hence $hx \in Hx$, and therefore, there exists some element $h_0 \in H$ such that $hx = h_0 x$. Rearranging this equality gives $h_0^{-1}hx = x$. Thus,

$$h_0^{-1}h \in \mathrm{Stab}_G(x).$$

By assumption B.3, we have

$$\mathrm{Stab}_G(x) \subseteq H.$$

Therefore $h_0^{-1}h \in H$. Since $h_0 \in H$ and $H$ is a subgroup, we have

$$h = h_0(h_0^{-1}h) \in H.$$

Thus every element $h \in \mathrm{supp}\, p_\theta(\cdot|x)$ lies in $H$, so

$$\mathrm{supp}\, p_\theta(\cdot|x) \subseteq H$$

for $q$-almost every $x \in S$.

It remains to show the marginal statement. The group-element distribution sampled by Algorithm 3 is obtained by marginalizing over data samples:

$$p_\theta(\cdot) = \int_{\mathcal{X}} p_\theta(\cdot|x)q(x)dx.$$

For $q$-almost every $x$, we have already shown that $\mathrm{supp}\, p_\theta(\cdot|x) \subseteq H$. A mixture of distributions supported on $H$ is also supported on $H$. Therefore,

$$\mathrm{supp}\, p_\theta \subseteq H.$$

$\square$

## C  Dataset

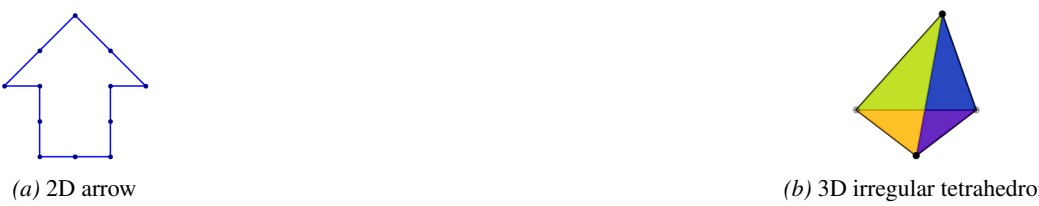

*(a)* 2D arrow                                    *(b)* 3D irregular tetrahedron

*Figure 7.* Canonical objects used for generating the 2D (a) and 3D (b) datasets

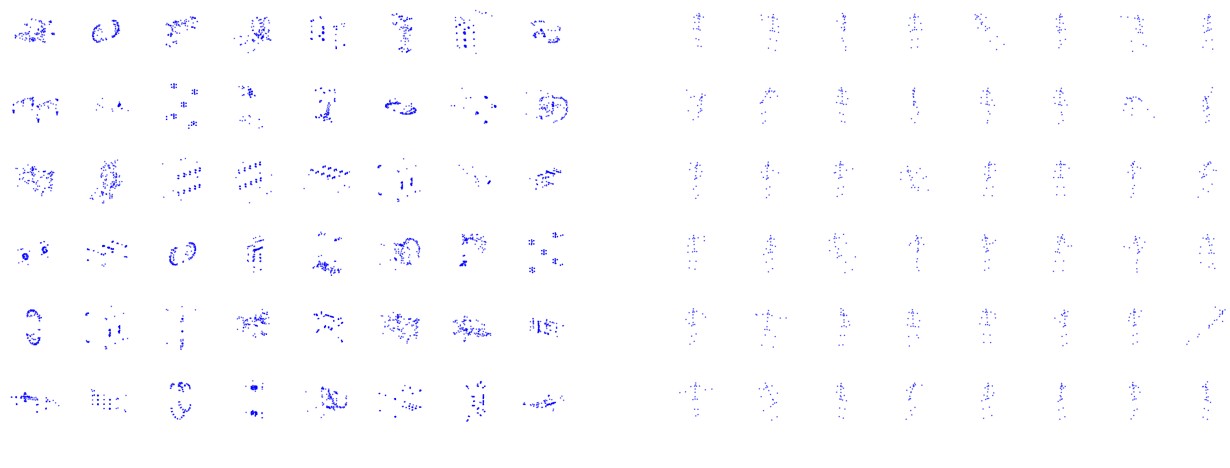

*(a)* ModelNet10 rotated under $\mathrm{Tet}$ group          *(b)* $3D$ skeletons from MI-Motion dataset

*Figure 8.* Samples from the ModelNet10 and MI-Motion datasets.

**2D arrow.**  For the 2D synthetic datasets, we use canonical arrow object as shown in Figure 7a. We generate datasets with discrete symmetries $C_4$.

**3D irregular tetrahedron.**  We use an irregular tetrahedron with no self-symmetries as shown in Figure 7b and generate datasets containing $\mathrm{Tet}, \mathrm{Oct}, \mathrm{Ico}$ and $\mathrm{SO}(2)$ symmetries.

**ModelNet10.**  To construct the rotated ModelNet10 dataset, we randomly select 10 objects from each category of ModelNet10 (Wu et al., 2015) as canonical shapes and uniformly sample 128 points from each surface to form point clouds. Each data point is then generated by independently sampling a canonical shape and a transformation from the target group and applying the transformation to the point cloud. Figure 8a. shows some samples of the ModelNet10 rotated under $\mathrm{Tet}$ group.

**MI-Motion.**  We use the MI-Motion dataset (Peng et al., 2023) to evaluate symmetry discovery on real-world human motion data. We randomly sample 500,000 3D skeletons from the dataset, where each skeleton consists of 20 joints. Due to the canonical alignment of motion trajectories in the world coordinate system, the extracted skeletons have global yaw orientations that are concentrated around four canonical directions. This induces an approximate $C_4$ symmetry in the data, which we aim to discover using `LieFlow`. Figure 8b. shows representative 3D skeleton samples from the MI-Motion dataset.

# D  Training Details

**Sampling Groups from Prior Distribution.**    For 2D arrow experiments, the prior known groups are $SO(2), GL(2, \mathbb{R})^+$. For $SO(2)$, we use a uniform distribution over the parameter $\theta \sim \mathcal{U}[-\pi, \pi]$ and generate the $2 \times 2$ rotation matrices. For the $GL(2)^+$ groups, we use two different sample strategies:

- **Lie algebra coefficients sampling**: We sample the coefficients $a, b, c, d \sim \mathcal{U}[-\pi, \pi]$ and construct the Lie algebra element as $A = \begin{bmatrix} a & b \\ c & d \end{bmatrix}$. Then we exponentiate it to obtain the transformation matrix $g = \exp(A)$.

- **Matrix composition based sampling**: We sample rotation angle $\theta \sim \mathcal{U}[-\pi, \pi]$, scaling factors $s_x, s_y \sim \mathcal{U}[0.5, 2.0]$ and shear factors $s_h \sim \mathcal{U}[-1, 1]$. Then we construct the transformation matrix as $g = \begin{bmatrix} \cos\theta & -\sin\theta \\ \sin\theta & \cos\theta \end{bmatrix} \begin{bmatrix} s_x & 0 \\ 0 & s_y \end{bmatrix} \begin{bmatrix} 1 & s_h \\ 0 & 1 \end{bmatrix}$.

Then, we reject any sampled transformation matrices with determinants outside $(0, 2]$ to ensure they belong to $GL(2)^+$ and make numerical stable.

For 3D experiments, we consider the prior group $SO(3)$ and use a uniform distribution over it by using Gaussian normalization over unit quaternions and transforming them into $3 \times 3$ matrices.

**Training setting for Synthetic data**    We use a time-conditioned MLP for flow matching that combines a sinusoidal time embedding module with a 5-layer MLP. The time embedding module uses sinusoidal positional encoding to transform the scalar time $t$ into a hidden dimensional vector, which is then projected to input dimension through a two-layer MLP with ReLU activations. The main network processes the concatenation of the flattened input and the time embedding through four hidden layers and an output layer with ReLU activations. The hidden dimension is set to $128$ for 2D arrow experiment and $512$ for 3D irregular tetrahedron experiment. We use the Adam optimizer with a learning rate of $3 \times 10^{-3}$ for both experiments.

**Training setting for ModelNet10**    For the ModelNet10 experiment, the prediction network adopted is a two-layer Transformer encoder with four attention heads and a hidden dimension of $256$. To condition the network on the time $t$, we use 64-dimensional sinusoidal time embeddings and concatenate them to the point features. The resulting per-point representations are aggregated via mean pooling to form a global feature. Then, the global feature is then passed through a three-layer MLP to predict a 3D Lie algebra element. AdamW optimizer is used with a learning rate of $1 \times 10^{-4}$ and weight decay of $1 \times 10^{-2}$.

**Time comparison**    We conduct training time comparisons between `LieFlow` and LieGAN on the Modelnet10 datasets. Both methods are trained on a single GeForce RTX 4090 GPU. Fixing batch size to $512$, for one epoch, `LieFlow` takes approximately $18$ seconds, while LieGAN takes around $50$ seconds.

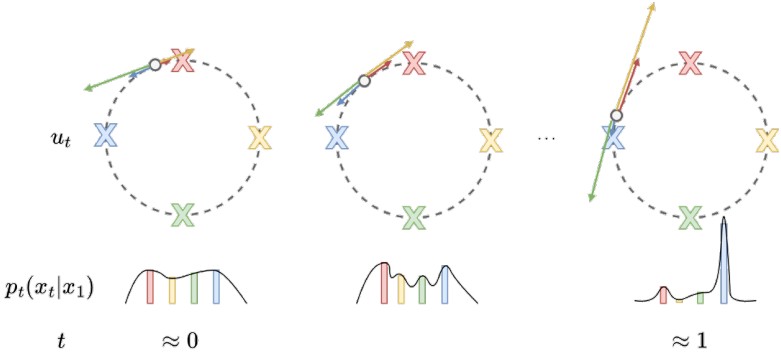

*Figure 9.* SO(2) to $C_4$ over group elements: target vector field and probability paths.

# E   Analysis on Last-Minute Mode Convergence

Given the challenges in learning the icosahedral group, we hypothesize that using flow matching for discovering discrete groups is in fact quite a difficult task. We illustrate with an even simpler scenario of flow matching directly on group elements (matrices), from SO(2) to $C_4$. As the target vector field $u_t$ and the probability path $p_t(x_t \mid x_1)$ are important quantities in our FM objective ((5)), we visualize them for a single sample in Figure 9.

Each target mode $x_1$ is colored differently and the velocities (group difference) to each $C_4$ group element are shown with the colored arrows. Near $t = 0$, the probability path $p_t$ is near uniform as $x_0$ is essentially noise and can be far away from $x_1$, i.e. there is still a lot of time $t$ for it to move to any mode. As $p_t$ is nearly uniform, the average of the velocities dominate and pulls $x_t$ close to the middle of the red and blue modes. Here, the vector field produces nearly 0 mean velocity, as it is equidistant to the red and blue modes, and equidistant to the green and yellow modes. The distribution $p_t$ becomes peakier towards the closer modes (red and blue), but still remains somewhat uniform. As such, the training target averages out close to 0. Near the end of training, when $t$ is close to 1, here the velocities become larger (due to the scaling factor $t$ inside the exponential) and $p_t$ now becomes more uni-modal, picking the closest mode (as there is little time left to go to the other modes, making them more unlikely). Thus, our task of finding a subgroup within a larger group is challenging precisely because the modes are "symmetric" (by definition of the subgroup).

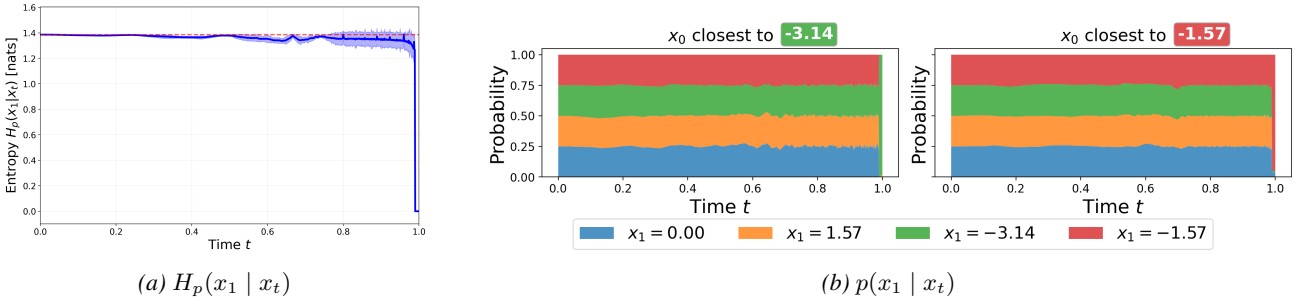

*(a) $H_p(x_1 \mid x_t)$*             *(b) $p(x_1 \mid x_t)$*

*Figure 10.* Left: The entropy of the posterior $p(x_1 \mid x_t)$ is generally uniform until $t \approx 1$.
Right: $x_0$ tends to converge to the closest mode/component (e.g., $x_0$ closest to $-\pi$ tends to converge to $-\pi$), but the posterior remains nearly uniform until $t \approx 1$.

**Flow Matching Directly on Group Elements.**   To verify our hypothesis, we perform flow matching directly on the group elements from SO(2) to $C_4$. The group elements can be parameterized by a scalar $\theta$, allowing us to analyze the behavior of the learned vector field more easily. Each data sample $x_1$ is a scalar from the set $\{-\pi, -\pi/2, 0, \pi/2\}$ and the source distribution is $q = \mathcal{U}[-\pi, \pi]$. We use the same network architecture as in the 2D experiments. We can further compute the reverse conditional or the posterior $p_t(x_1 \mid x_t)$ that can help verify two things: 1) that $x_t$ remain nearly stationary until $t$ approaches 1, i.e. $p(x_l \mid x_t)$ remains nearly uniform, and 2) $x_t$ chooses the closest group element $x_1$. The exact details of how $p_t(x_1 \mid x_t)$ is computed is given in Appendix E.2.

Figure 12 in Appendix E.1 visualizes the progression of $x_t$ over time for 100 samples, showing that $C_4$ symmetry is clearly identified. Figures 10a plots the entropy of the posterior, and shows that even in this simple scenario, the entropy remains near the maximum (uniform distribution) until $t$ approaches 1. Figure 10b visualizes the posterior $p(x_1 \mid x_t)$ averaged over

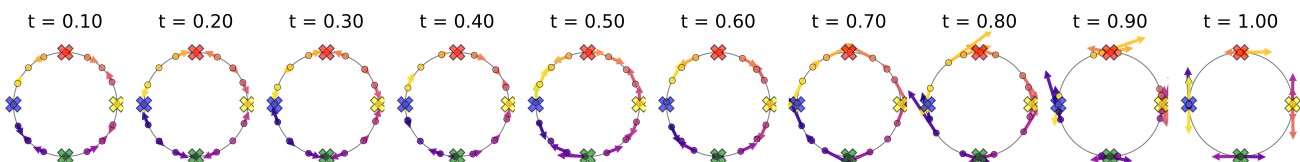

*Figure 11.* SO(2) to $C_4$ group elements: Evolution of $x_0$'s and their velocities over time.

1,000 samples and separated into 4 classes, depending on the original $x_0$ and its nearest mode in $\{-\pi, -\pi/2, 0, \pi/2\}$ (only two shown, other classes are similar, see Figures 13b in Appendix E.1). For all 4 classes, we observe that $p(x_1 \mid x_t)$ remains nearly uniform until around $t = 0.9$, when it becomes uni-modal to the closest mode/component, e.g., for all $x_0$ that were closest to the mode $\mu = -\pi$, then the posterior $p(x_1 \mid x_t)$ converges to $\mu$. This shows both that the learned vector field remains nearly 0 for most timesteps, only converges when little time is left, and converges to the closest group element.

Note that this differs from the standard FM setting, where the flow is defined on a high-dimensional space. Bertrand et al. (2026) find that for high-dimensional flow matching with images, the entropy of the posterior drops at small times, transitioning from a stochastic phase that enables generalization to a non-stochastic phase that matches the target modes. In our low-dimensional setting of symmetry discovery, we find the opposite occurs: the entropy stays near uniform until large $t$, where it suddenly converges. We term this phenomenon "last-minute mode convergence".

Figure 11 shows the evolution of a uniform grid over SO(2) over time. As we hypothesized, we can see from the velocity arrows that particles move slowly closer to the midpoint between two modes ($t = 0$ to $t = 0.60$) but the velocities are relatively small. From $t = 0.70$ onwards, we can see that the shift towards the closest modes and the velocities become increasingly large until $t = 1$. Figures 14 (Appendix E.1) shows a clearer picture of how the velocity evolves over time. From $t = 0$ to $t = 0.6$, we see that the $x_t$'s are nudged towards the midpoints between modes. Interestingly, we see some oscillating behavior where the vector field flips sign, suggesting that the net velocity hovers close to 0 with no defined pattern. From $t = 0.8$ onwards, we see that points $x_t$ are pushed to the closest $x_1$ mode, with increasing velocities. With near zero net velocity until $t = 0.6$, these plots also support the observation of "last-minute mode convergence".

### E.1 Flow Matching on Group Elements

This section contains additional visualizations for the flow matching experiment on group elements from SO(2) to $C_4$ as described in Section E.

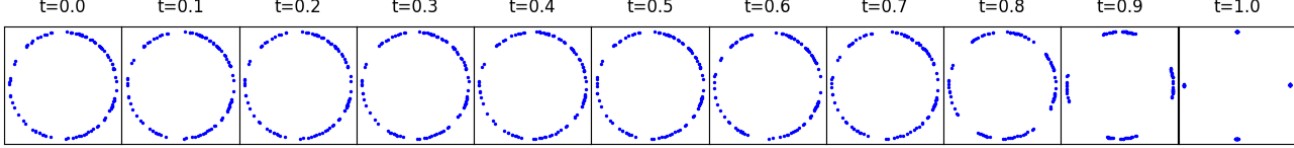

*Figure 12.* SO(2) to $C_4$ group elements: Visualization of 100 samples over time.

Figures 12 shows the time progression of $x_t$ when generating 100 samples from $t = 0$ to $t = 1$. We can see that the samples converge to the closest group element in the orbit.

Figures 13b in Figures 13 shows the posterior $p(x_1 \mid x_t)$ for all 4 classes, depending on the original $x_0$ and its nearest mode in $\{-\pi, -\pi/2, 0, \pi/2\}$. We can see that for all 4 classes, the posterior $p(x_1 \mid x_t)$ remains nearly uniform when $t$ closes to 1.

Figures 13 visualizes the posterior $p(x_1 \mid x_t)$, divided into 4 buckets depending on the initial position $x_0$. We see that the posterior remains nearly uniform until $t = 0.95$, and then $x_t$ converges to the mode that was the closest when $t = 0$.

Figures 14 visualizes how the velocities change over time. Until $t = 0.6$, the velocities are close to 0 but slightly push the $x_t$'s towards the midpoints between modes. From $t = 0.8$ onwards, we see that points $x_t$ are pushed to the closest $x_1$ mode, with increasing velocities.

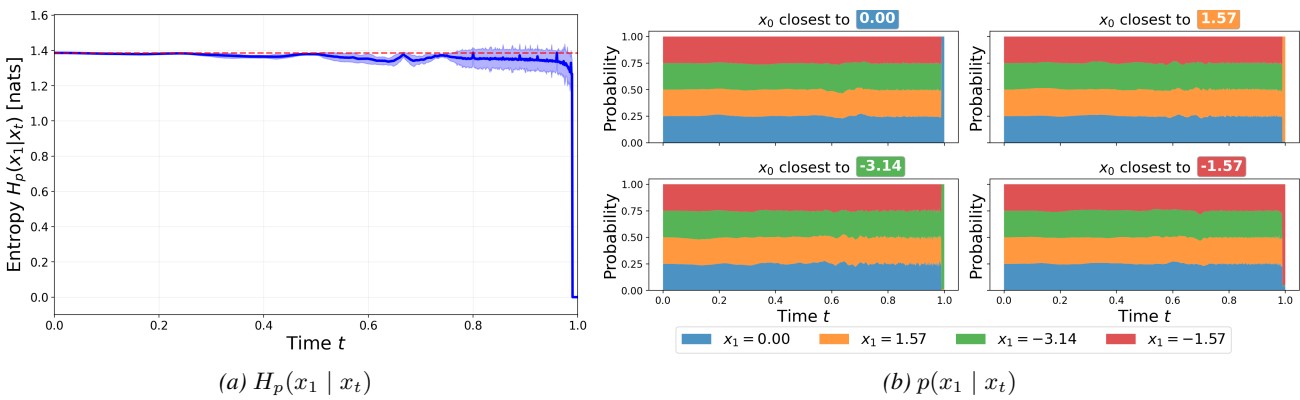

*(a) $H_p(x_1 \mid x_t)$*          *(b) $p(x_1 \mid x_t)$*

*Figure 13.* Visualizations of the posterior $p(x_1 \mid x_t)$. The left graph shows the entropy of the posterior is generally uniform until $t \approx 1$. The right graph visualizes $p(x_1 \mid x_t)$ for each target mode and demonstrates that the generated samples, depending on the position of $x_0$, converge to the closest mode.

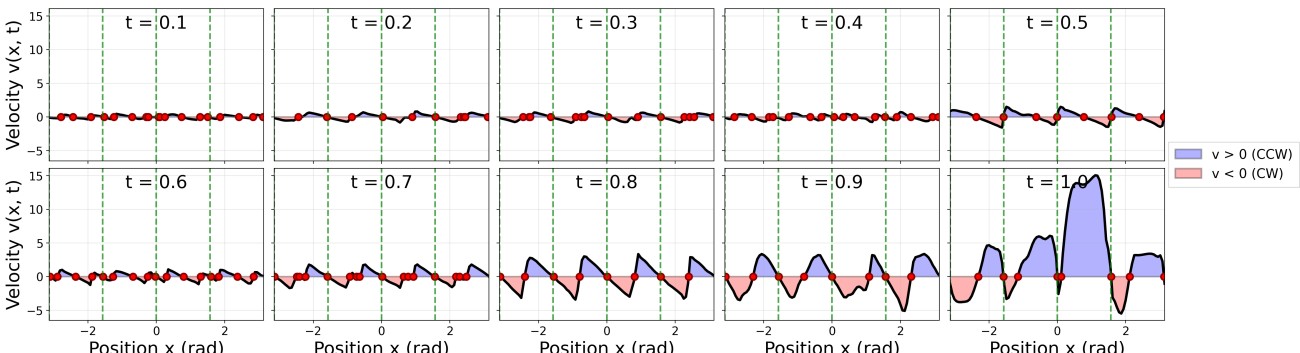

*Figure 14.* SO(2) to $C_4$ group elements: velocity phase portraits over time.

## E.2 Derivation of Posterior Computation

To compute the posterior, we use Bayes' rule as

$$p(x_1|x_t) = \frac{p(x_t|x_1)p(x_1)}{p_t(x_t)}, \tag{11}$$

where $p(x_t \mid x_1)$ is the likelihood and $p(x_1)$ is the data distribution. Since flow matching learns a velocity field rather than explicit conditional distributions, we cannot directly compute the likelihood and need to invert the flow and integrate the continuity equation.

We first sample initial points $x_0$ from a uniform prior over $[-\pi, \pi)$ and forward simulate using the learned velocity field $v_\theta$ to obtain the trajectories $x_t$ and their log probabilities through the continuity equation. To evaluate the likelihood $p(x_t \mid x_1)$ for each candidate $x_1$, we invert the linear interpolation $x_t = (1-t)x_0 + tx_1$ to find $x_0 = (x_t - tx_1)/(1-t)$, then integrate the divergence along the trajectory from this inverted $x_0$. Special handling is required at the boundaries: at $t = 0$, the posterior equals the prior $p(x_1)$ (which we know in this simple scenario) due to independence, while near $t = 1$, we use a Gaussian approximation to avoid numerical instability from division by $(1 - t)$.

---

**Algorithm 4** Compute Posterior $p(x_1 \mid x_t)$ for Flow Matching

---

**Require:** Velocity field $v_\theta(x, t)$, prior $p_0 \sim \mathcal{U}[-\pi, \pi]$, target $p_1(x_1) = \sum_{k=1}^{K} \frac{1}{K}\delta(x_1 - \mu^{(k)})$, time steps $T$
  1: **Initialize:** Sample $\{x_0\}^N \sim p_0$
  2:
  3: **Forward Simulation:**
  4: **for** $t \in \{0, \frac{1}{T}, \ldots, 1\}$ **do**
  5:   $x_t \leftarrow \text{ODESolve}(v_\theta, x_0, [0, t])$          ▷ *Integrate velocity field*
  6:   $\log p_t(x_t) \leftarrow \log p_0(x_0) - \int_0^t \nabla \cdot v_\theta(x_s, s)ds$          ▷ *Continuity equation*
  7: **end for**
  8:
  9: **Posterior Computation:**
 10: **for** $t \in \{0, \frac{1}{T}, \ldots, 1\}$ **do**
 11:   **if** $t = 0$ **then**
 12:     $p(x_1 \mid x_0) \leftarrow p_1(x_1)$          ▷ *Independent at $t = 0$*
 13:   **else if** $t \approx 1$ **then**
 14:     **for** $k = 1$ to $K$ **do**
 15:       $\log p(x_t \mid x_1^{(k)}) \leftarrow -\frac{\|x_t - x_1^{(k)}\|^2}{2\sigma^2}$, where $\sigma = 0.1$          ▷ *Gaussian Approximation*
 16:     **end for**
 17:   **else**
 18:     **for** $k = 1$ to $K$ **do**
 19:       $x_0^{(k)} \leftarrow \frac{x_t - t \cdot x_1^{(k)}}{1 - t}$          ▷ *Invert linear interpolation*
 20:       Forward simulate path from $x_0^{(k)}$ to time $t$ using $v_\theta$ and continuity equation
 21:       $\log p(x_t \mid x_1^{(k)}) \leftarrow \log p_0(x_0^{(k)}) - \int_0^t \nabla \cdot v_\theta(x_s, s)ds$
 22:     **end for**
 23:   **end if**
 24:   **for** $k = 1$ to $K$ **do**
 25:     $\log p(x_1^{(k)} \mid x_t) \leftarrow \log p(x_t \mid x_1^{(k)}) + \log p_1(x_1^{(k)})$          ▷ *Bayes' Rule*
 26:   **end for**
 27:
 28:   $p(x_1^{(k)} \mid x_t) \leftarrow \frac{\exp(\log p(x_1^{(k)}|x_t))}{\sum_k \exp(\log p(x_1^{(k)}|x_t))}$          ▷ *Normalize over $K$ components*
 29: **end for**
 30:
 31: **Return** Posterior $\{p(x_1 \mid x_t)\}$ for all time steps

---

# F   Power time schedule

Figure 15 shows the density of the power distribution for different skewness values.

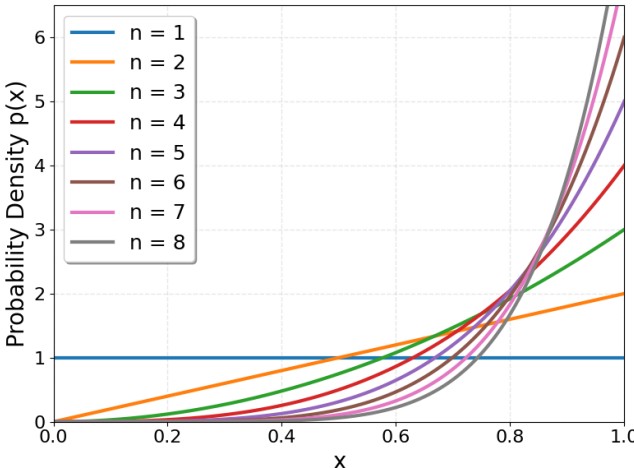

*Figure 15.* Power distribution density

Figures 16 compares the time progression of $x_t$ between without and with power time schedule for $SO(3)$ to Ico dataset. Figure 16(a) shows that without power time schedule, the samples remain nearly stationary until $t$ approaches 1, while in Figure 16(b), with power time schedule, the samples start moving towards the target modes earlier, even at small $t$ values. This indicates that the power time schedule helps to alleviate the last-minute mode convergence issue discussed in Section 5.4 and Appendix E.

## F.1   Ablation study on skewness

We perform an ablation study on the skewness parameter of the power time schedule for $SO(3)$ to Ico setting. Table 4 reports the Wasserstein-1 distance between the generated transformations and the ground truth group elements for different skewness values. Figure 17 visualizes the training curves for different skewness values. From the results, we observe that increasing the skewness of the power time schedule leads to better performance and faster convergence. Since the improvement saturates around skewness 5, we use this value for all 3D experiments in the main paper.

*Table 4.* Ablation study on skewness of power time schedule for $SO(3)$ to Ico setting. Wasserstein-1 distance between the generated transformations and the ground truth group elements are reported. Lower is better.

| skewness | 1 (uniform) | 2 | 3 | 4 | 5 |
|---|---|---|---|---|---|
| | $0.470 \pm 0.071$ | $0.157 \pm 0.019$ | $0.123 \pm 0.017$ | $0.109 \pm 0.010$ | $0.104 \pm 0.008$ |

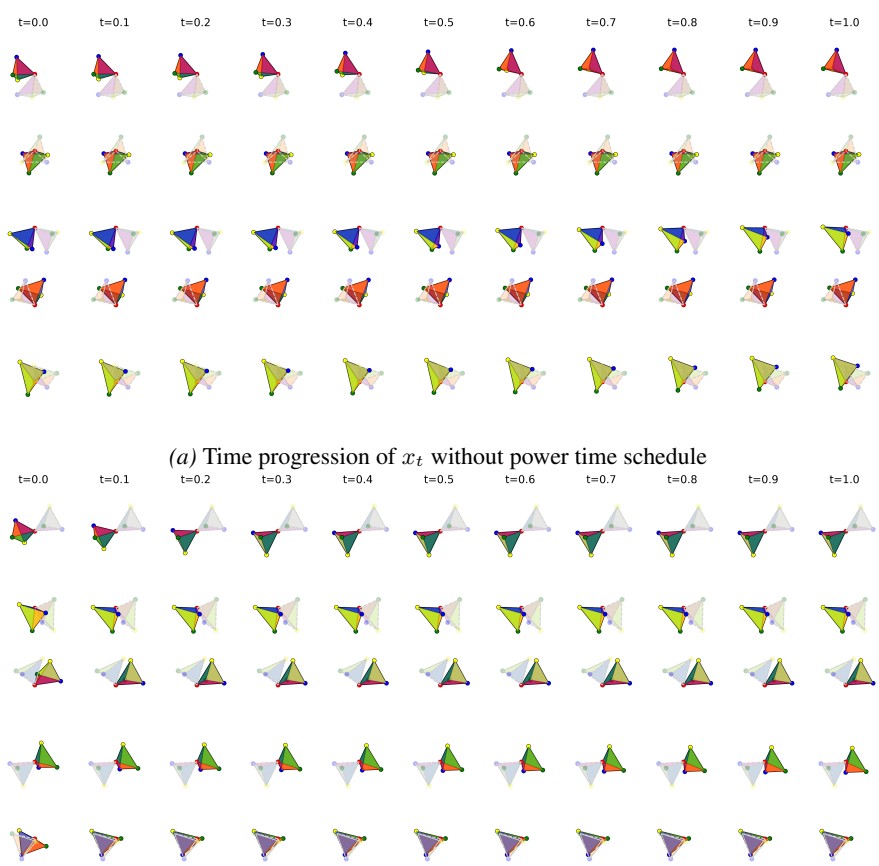

*(a)* Time progression of $x_t$ without power time schedule

*(b)* Time progression of $x_t$ with power time schedule

*Figure 16.* Test progression comparison between without and with power time schedule for $\mathrm{SO}(3)$ to Ico dataset.

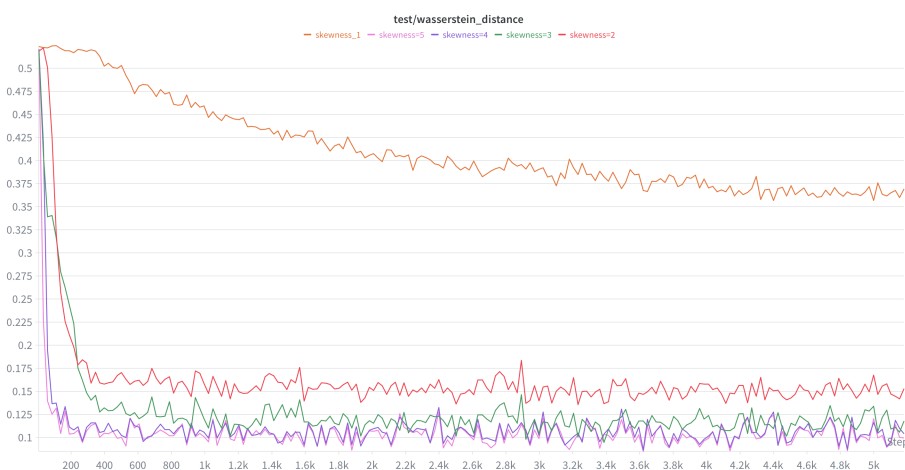

*Figure 17.* Ablation study on the skewness of power time schedule for $\mathrm{SO}(3)$ to Ico setting. $y$ axis denote the Wasserstein-1 distance between the generated transformations and the ground truth group elements, and $x$ axis denote training epoch.

# G   Additional Results

This section contains additional visualization results.

## G.1   Discovering symmetry with non-trivial stabilizer

We further test LieFlow on a rectangle, which has a non-trivial discrete self-symmetry $C_2$. This setting addresses the stabilizer issue discussed in Appendix A: although the stabilizer is non-trivial, it is finite and hence has zero Lie algebra, $\mathrm{Lie}(C_2) = \{0\}$, so it introduces no infinitesimal degeneracy in the Lie algebra target. Using $\mathrm{SO}(2)$ as the hypothesis group, we generate a dataset from $C_4$ rotations of the rectangle. Figure 18 shows that LieFlow recovers four peaks corresponding to the elements of $C_4$. This demonstrates that discrete self-symmetries do not prevent symmetry discovery.

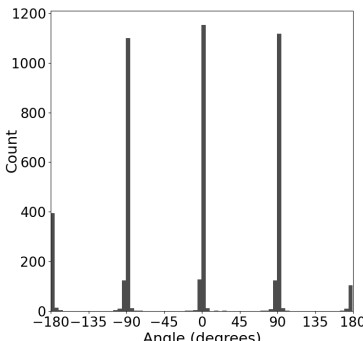

Figure 18. $\mathrm{SO}(2)$ to $C_4$ rectangle experiment: histogram of the generated transformations. The four peaks correspond to the four elements of $C_4$.

## G.2   2D arrow

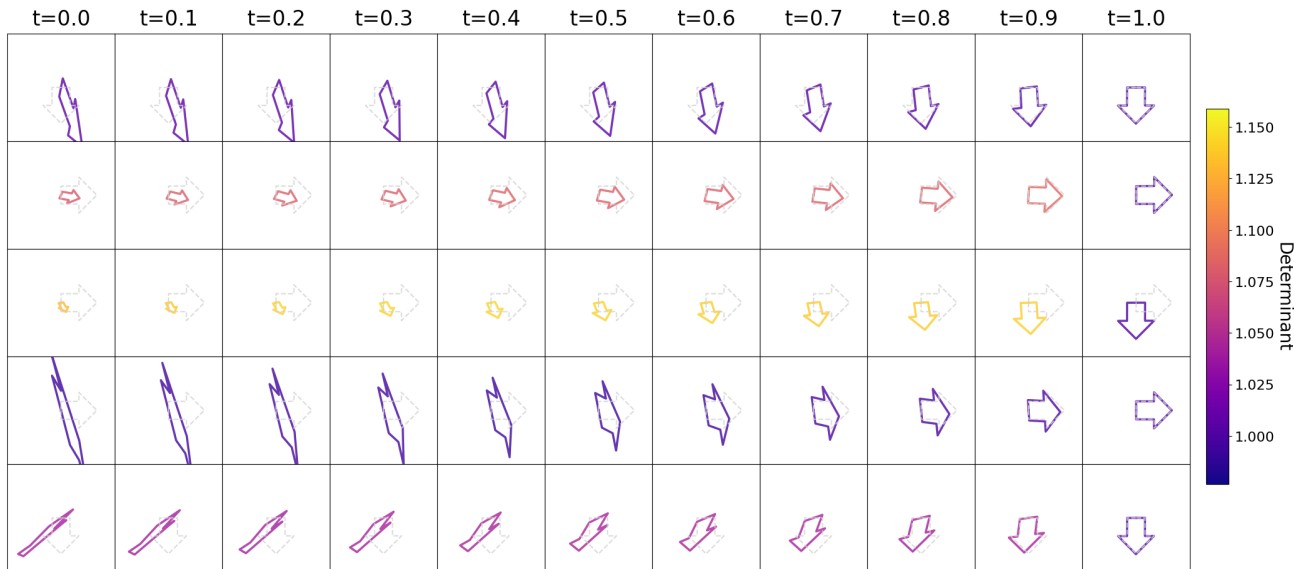

Figure 19. $\mathrm{GL}(2)^+$ to $C_4$: Time progression of $x_t$ when generating 5 samples over 20 steps. The gray arrow shows the original $x_1$ and the color represents the determinant of the generated transformation matrix.

Figures 19 shows the time progression of $x_t$ when generating 5 samples over 20 steps for $\mathrm{GL}(2)^+$ to $C_4$. The gray arrow shows the original $x_1$ and the color represents the determinant of the generated transformation matrix. We can see that the transformed point clouds converge to the closest group element in the orbit.

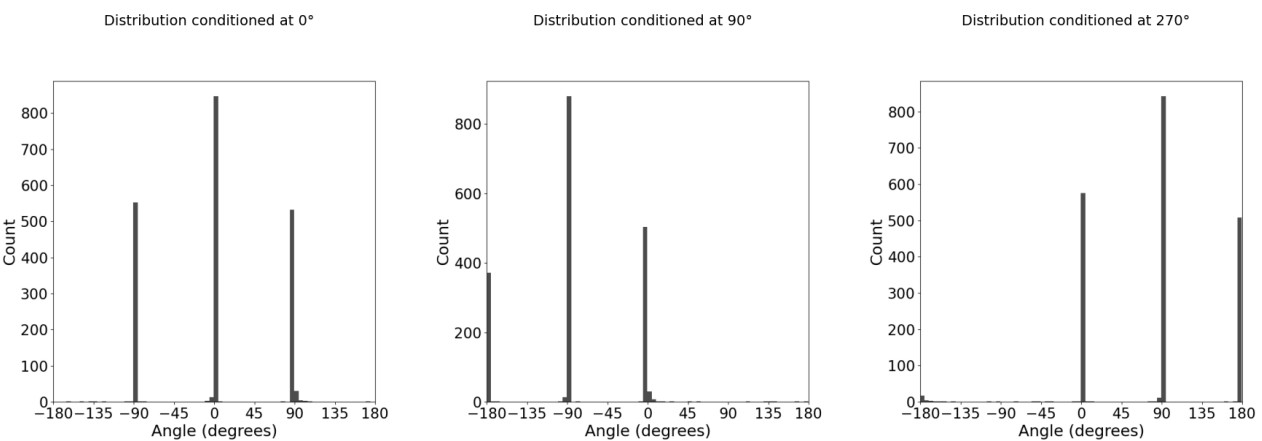

*Figure 20.* Histograms of the conditional angle distribution of the three arrows with $0°$, $90°$ and $270°$ rotation.

Figure 20 shows the results in partially observed symmetry setting. By fixing $x_1$ and run multiple times of Algorithm 3, we can obtain the distribution of the generated transformations conditioned on each sample. We visualize the histogram of the angle distribution for three different $x_1$ with $0°$, $90°$ and $270°$ rotation, and the results align with our expectation discussed in section 4.3.

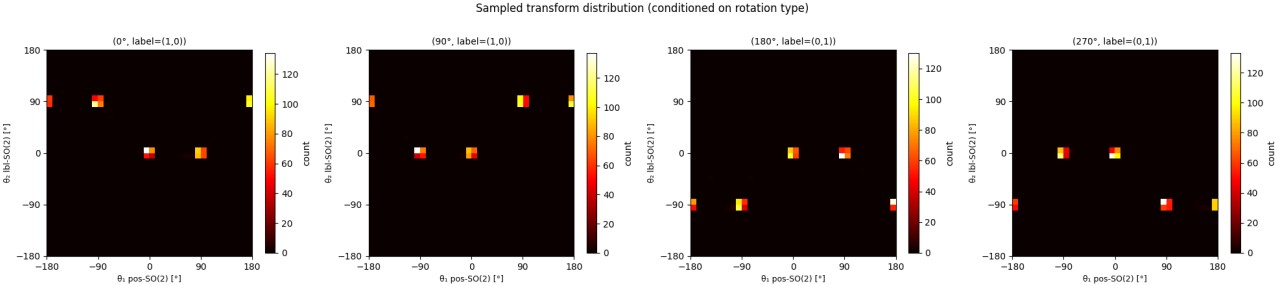

*Figure 21.* 2D Histograms of the conditional angle distribution. Lighter color indicates higher density. The $x$ axis is the rotation angle on input while the $y$ axis is the rotation angle on label.

Figure 21 shows the results in functional partial symmetry setting. We such setting as discovering partially observed symmetry in the joint distribution $p(x, y)$. We visualize the 2D histogram of the angle distribution for each input-label pair $(x, y)$, and label-preserving and label-changing transformations for each sample can be infered from these histograms. For example, for the input-label pair with $90°$ rotation, we can see that if $90°$ act on such input, the label will also rotate $90°$ (from $(1, 0)$ to $(0, 1)$), while if $-90°$ act on such input, the label will remain the same.

### G.3 3D irregular tetrahedron

Figures 22 visualizes the trajectories of the centroids over time for 100 samples, and PCA is performed to project them onto the 2D plane. We can see that our model seems to learn Tet, Oct ,Ico and $\mathrm{SO}(2)$ elements, creating some visible clusters.

We also show the intermediate $x_t$ during generation in Figure 23, Figure 24, Figure 25 and Figure 26 for Tet, Oct, Ico and $\mathrm{SO}(2)$ respectively. For the Tet, Oct, and Ico, we see that the transformed point clouds converge to the closest group element in the orbit instead of the original $x_1$, which shows that the model has learned the symmetry structure of the data. For the $\mathrm{SO}(2)$ group, we can see that the model produces rotations to turn one node of the tetrahedron to be aligned with the $z$-axis and one triangle face to be in the $xy$-plane, which corresponds to $\mathrm{SO}(2)$ symmetries around the $z$-axis. Table 5 reports the quantitative comparison between `LieFlow` and LieGAN for all 5 target groups. We can see that `LieFlow` significantly outperforms LieGAN in discovering discrete group.

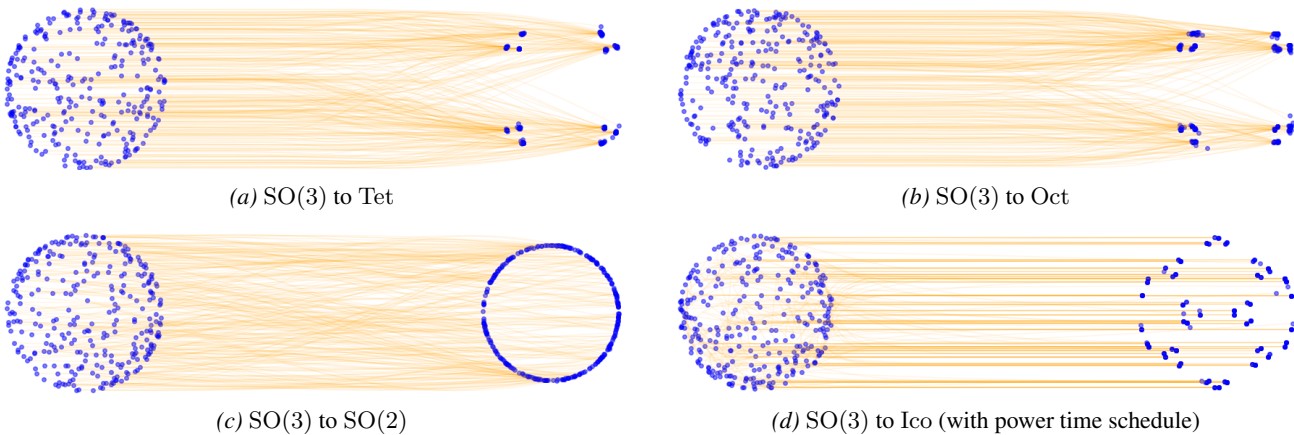

*(a)* SO(3) to Tet

*(b)* SO(3) to Oct

*(c)* SO(3) to SO(2)

*(d)* SO(3) to Ico (with power time schedule)

*Figure 22.* A 2D PCA visualization of trajectories from t = 0 (left) to t = 1 (right) of the centroids of the transformed objects over 100 samples.

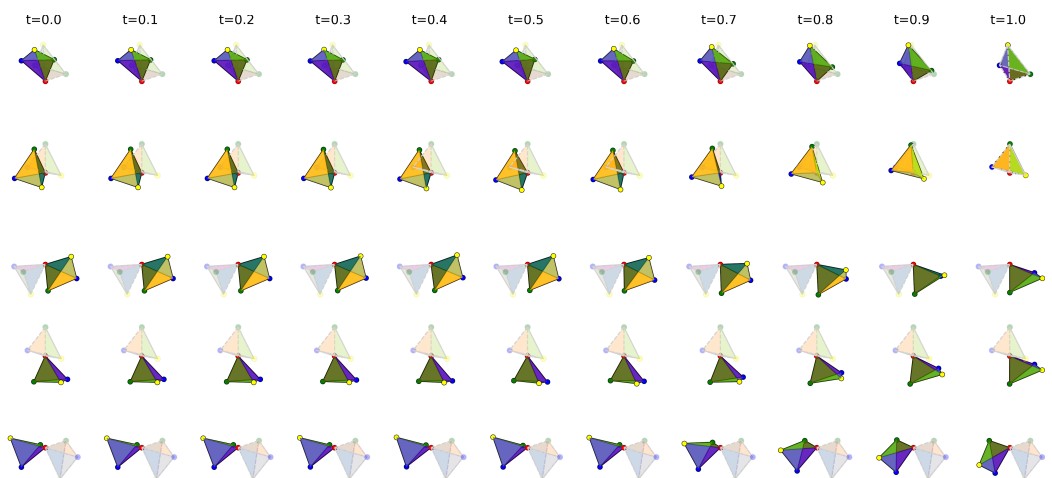

*Figure 23.* SO(3) to Tet: time progression of $x_t$ when generating 5 samples over 100 steps. The transparent tetrahedron indicates the original $x_1$.

## G.4   ModelNet10

Visualization results for rotated ModelNet10 are shown in Figures 27. Visualizations of robust analysis are shown in Figures 28. The generated group elements are clustered around the ground truth group elements for all target groups, which means that our model successfully discovered the symmetries in the rotated ModelNet10 dataset. Note that for the Ico group, the situation here is even worse than the irregular tetrahedron case without power time schedule, as the model fails to discover any symmetry due to the increased complexity of the dataset. However, with the power time schedule, the model still is able to discover the Ico symmetries, even with some noises that break the exact symmetry. This shows the effectiveness and robustness of our model combined with the power time schedule in discovering complex symmetries.

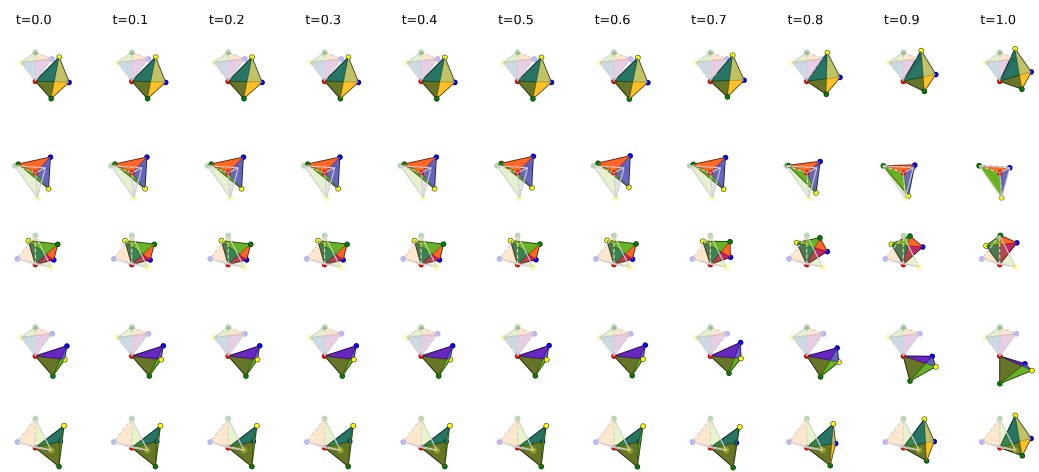

*Figure 24.* SO(3) to Oct: time progression of $x_t$ when generating 5 samples over 100 steps. The transparent tetrahedron indicates the original $x_1$.

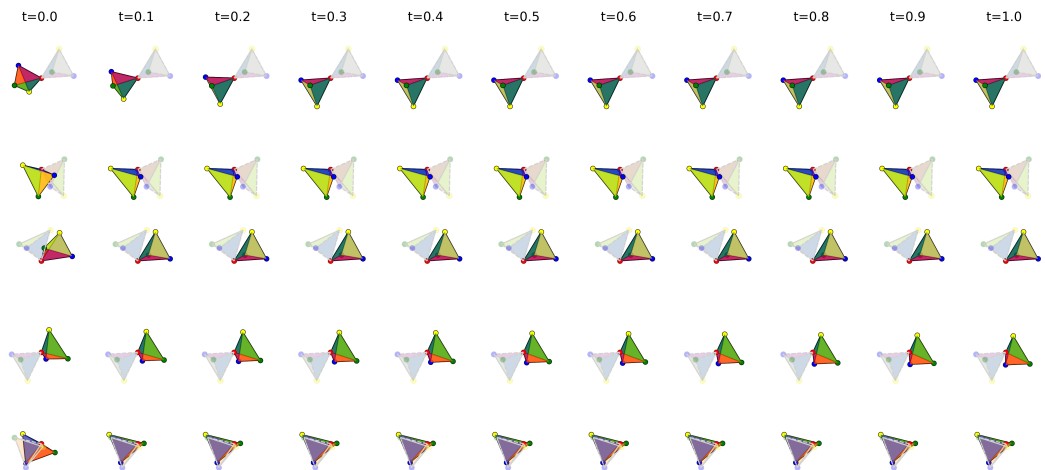

*Figure 25.* SO(3) to Ico: time progression of $x_t$ when generating 5 samples over 100 steps. The transparent tetrahedron indicates the original $x_1$. Training with power time schedule.

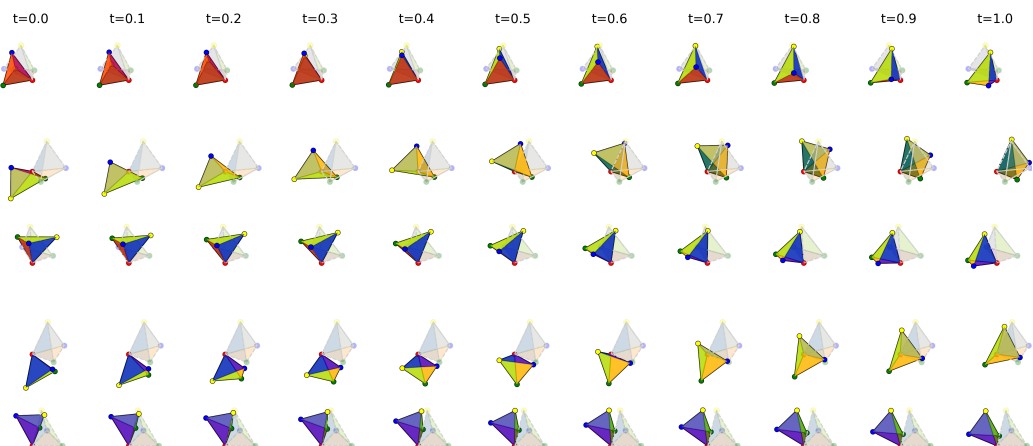

*Figure 26.* SO(3) to SO(2): time progression of $x_t$ when generating 5 samples over 100 steps. The transparent tetrahedron indicates the original $x_1$.

*Table 5.* Wasserstein-1 distance and relative improvement over random sampling in irregular tetrahedron experiments. We report $W_1$ (RI%), where lower $W_1$ and higher RI are better. $\mathrm{SO}(3) \to \mathrm{SO}(2)$(a) corresponds to rotations around $z$ axis and $\mathrm{SO}(3) \to \mathrm{SO}(2)$(b) corresponds to rotations around $(0, 1/2, -\sqrt{3}/2)$ axis. Results are averaged over 3 random seeds and the standard deviation is shown for all methods.

| Method | $\mathrm{SO}(3) \to \mathrm{Tet}$ | $\mathrm{SO}(3) \to \mathrm{Oct}$ | $\mathrm{SO}(3) \to \mathrm{SO}(2)$(a) | $\mathrm{SO}(3) \to \mathrm{SO}(2)$(b) | $\mathrm{SO}(3) \to \mathrm{Ico}$ |
|---|---|---|---|---|---|
| LieGAN | $1.263 \pm 0.036 \, (-40.2\%)$ | $1.226 \pm 0.017 \, (-71.1\%)$ | $0.032 \pm 0.010 \, (98.0\%)$ | $\mathbf{0.029 \pm 0.008} \, (\mathbf{98.2}\%)$ | $1.198 \pm 0.064 \, (-131.0\%)$ |
| LieFlow(Ours) | $\mathbf{0.066 \pm 0.012} \, (\mathbf{92.7}\%)$ | $\mathbf{0.073 \pm 0.010} \, (\mathbf{89.8}\%)$ | $\mathbf{0.027 \pm 0.009} \, (\mathbf{98.3}\%)$ | $0.032 \pm 0.015 \, (98.0\%)$ | $\mathbf{0.104 \pm 0.008} \, (\mathbf{80.0}\%)$ |

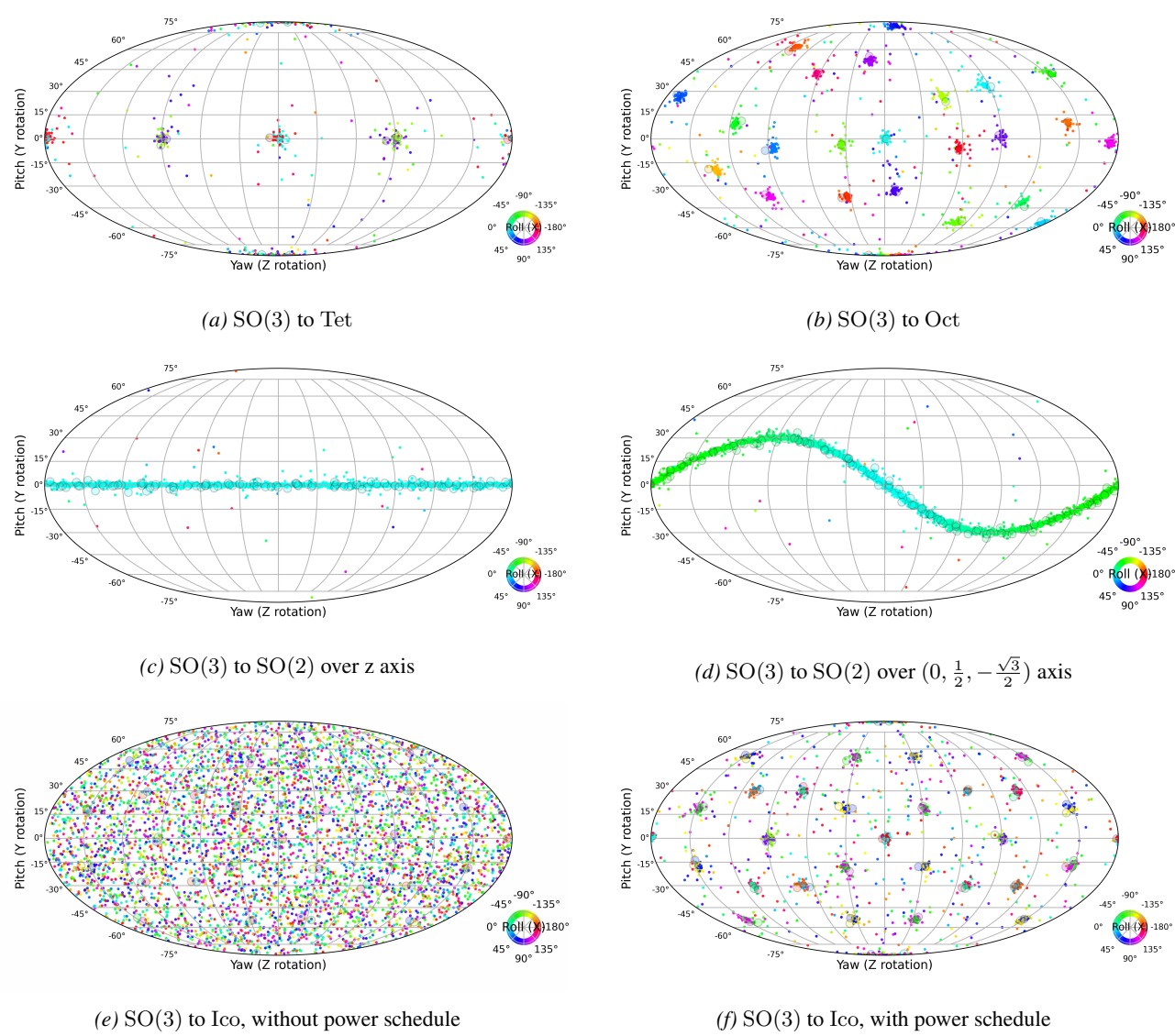

*(a)* $\mathrm{SO}(3)$ to Tet

*(b)* $\mathrm{SO}(3)$ to Oct

*(c)* $\mathrm{SO}(3)$ to $\mathrm{SO}(2)$ over z axis

*(d)* $\mathrm{SO}(3)$ to $\mathrm{SO}(2)$ over $(0, \frac{1}{2}, -\frac{\sqrt{3}}{2})$ axis

*(e)* $\mathrm{SO}(3)$ to Ico, without power schedule

*(f)* $\mathrm{SO}(3)$ to Ico, with power schedule

*Figure 27.* Modelnet10 results: Visualization of 5,000 generated elements of $\mathrm{SO}(3)$ by converting them to Euler angles. The first two angles are represented spatially on the sphere using Mollweide projection and the color represents the third angle. The elements are canonicalized by the original random transformation and the ground truth elements of the target group are shown in circles with gray borders. The points are jittered with uniformly random noise to prevent overlapping.

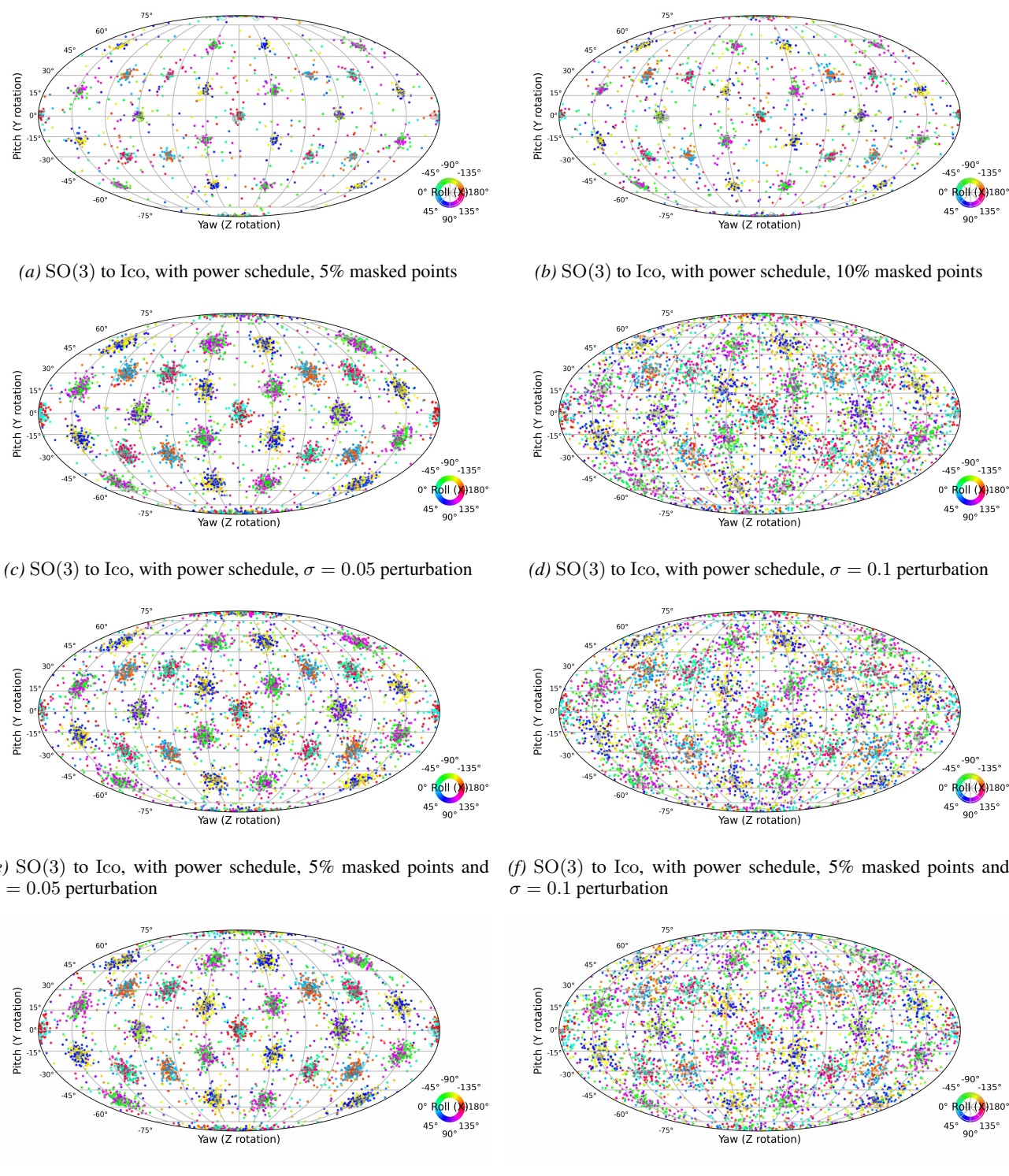

*(a)* SO(3) to Ico, with power schedule, 5% masked points

*(b)* SO(3) to Ico, with power schedule, 10% masked points

*(c)* SO(3) to Ico, with power schedule, $\sigma = 0.05$ perturbation

*(d)* SO(3) to Ico, with power schedule, $\sigma = 0.1$ perturbation

*(e)* SO(3) to Ico, with power schedule, 5% masked points and $\sigma = 0.05$ perturbation

*(f)* SO(3) to Ico, with power schedule, 5% masked points and $\sigma = 0.1$ perturbation

*(g)* SO(3) to Ico, with power schedule, 10% masked points and $\sigma = 0.05$ perturbation

*(h)* SO(3) to Ico, with power schedule, 10% masked points and $\sigma = 0.1$ perturbation

*Figure 28.* Visualizations of robust analysis: Visualization of 5,000 generated elements of SO(3) by converting them to Euler angles. The first two angles are represented spatially on the sphere using Mollweide projection and the color represents the third angle. The elements are canonicalized by the original random transformation and the ground truth elements of the target group are shown in circles with gray borders. The points are jittered with uniformly random noise to prevent overlapping.

