# OpenReview forum: "Discovering Symmetry Groups with Flow Matching"
_ICML.cc/2026/Conference — ICML 2026 regular_

### Official Review · Reviewer_5LQj · 2026-03-09

**Soundness:** 2
**Presentation:** 3
**Significance:** 3
**Originality:** 3
**Overall Recommendation:** 4
**Confidence:** 4

**Summary:**

The paper proposes LieFlow, a framework for discovering symmetry groups by learning probability distributions over Lie group elements using flow matching. Instead of learning Lie algebra generators with a fixed prior over coefficients, the method operates directly on a hypothesis Lie group \(G\), training a conditional flow that maps prior samples in \(G\) to transformations consistent with the data distribution. The support of the learned distribution reveals the underlying symmetry subgroup \(H \subseteq G\), allowing a unified treatment of continuous and discrete, as well as partially observed, symmetries. Experiments on synthetic 2D and 3D point clouds and on rotated ModelNet10 evaluate recovery of various symmetry groups, comparing mainly to LieGAN using Wasserstein-1 distance on group elements and visualizations of learned transformations. A special non-uniform time sampling schedule is introduced to stabilize learning of discrete symmetries.

**Compliance With Llm Reviewing Policy:**

Affirmed.

**Final Justification:**

The authors have provided thorough and detailed responses to my concerns, including new experiments and clarifications that address the points I raised. I will keep my scores but rise the confidence.

**Key Questions For Authors:**

**1. Sensitivity to the choice of hypothesis group.**
The authors should clarify how sensitive LieFlow is to the choice of hypothesis group when it does not contain the true symmetry group. For example, what happens if the model is trained on a dataset with Ico symmetry while using \(SO(2)\) as the hypothesis group, or when using an incorrect non-compact group to model a compact symmetry? Some diagnostic experiments or theoretical discussion would help clarify the practical risks of mis-specifying \(G\).

**2. Clarification of the Wasserstein-1 distance metric.**
The authors should provide more detail on the distance metric used in the Wasserstein-1 computation on Lie groups. In particular, for \(SO(3)\), how is \(d_G\) computed between two rotations? It would also be helpful to know whether alternative metrics have been tested. For example, do the conclusions regarding relative performance versus LieGAN remain consistent when using angle-based distances versus Frobenius-norm-based distances?

**Limitations:**

No.

The paper includes an Impact Statement but it is very brief and does not meaningfully discuss potential societal impacts or limitations.

The authors could improve this section by briefly discussing possible implications of automatic symmetry discovery, such as risks of incorrect symmetry identification affecting scientific interpretation or downstream models.

**Strengths And Weaknesses:**

## Strengths

**1. Conceptually clean formulation of symmetry discovery as distribution learning on groups.**
The paper proposes to learn a distribution over a hypothesis Lie group whose support converges to the true symmetry subgroup (Section 4.1). This framing cleanly separates the choice of hypothesis group from the identification of actual symmetries, and avoids imposing a fixed analytic form for the transformation distribution. The framework naturally accommodates both continuous and discrete symmetries, which is demonstrated empirically on SO(2), GL(2)^+, and SO(3) with various discrete subgroups .

**2. Technically sound Lie-group-aware flow matching formulation.**
The paper carefully derives a flow-matching objective defined on Lie groups. Orbit-respecting interpolants (x_t = exp(tA)x_0) (Eq. 4) and a Lie-algebra parameterized vector field (Eq. 5) allow the model to learn flows directly within the group structure. The derivation via pushforwards and the orbit–stabilizer discussion provides a convincing geometric justification for the method.

**3. Insightful analysis and practical solution for discrete symmetry learning.**
The paper provides a clear analysis of why discrete subgroup discovery is challenging for flow matching: the posterior over modes remains nearly uniform until late times, leading to vanishing effective velocity fields. The proposed power time-sampling schedule focuses training on late-time regions where gradients are informative, leading to significant improvements over LieGAN in Wasserstein distance across several settings. Visualizations throughout the paper clearly illustrate the learned symmetry structures.


## Weaknesses

**1. Insufficient comparison with related methods.**
The quantitative evaluation compares primarily against LieGAN. While this is a reasonable baseline, several closely related symmetry discovery approaches (e.g., Allingham et al., 2024; SymmetryGAN) are not included experimentally, which makes it difficult to fully assess the relative performance of the method.

**2. Limited analysis of robustness and practical impact.**
Although the method is motivated by the potential benefits of symmetry discovery for equivariant learning and sample efficiency, no downstream task demonstrates these advantages. Additionally, sensitivity to prior parameterization (e.g., the large performance difference between GL(2)^+ sampling schemes in Table 1) suggests the method may depend on careful prior design, but this is not systematically analyzed.

**3. Minimal evaluation of partial or approximate symmetries.**
The experiments on partial symmetries are restricted to a toy example involving missing elements of C4. More realistic scenarios—such as input-dependent symmetries, approximate invariances, or symmetry breaking in noisy real-world data—are not explored. Statistical uncertainty is also not reported, as results are given without standard deviations across runs.

---

> ### Author Rebuttal · Authors · 2026-03-31
>
> We thank the reviewer for their thoughtful review and helpful feedback.
> ## Weaknesses:
> > W1: ``Insufficient comparison with related methods’’
>
> We added Augerino (Benton et al., 2020) and SGM (Allingham et al., 2024) on ModelNet10. For Augerino, we assign the same label to all points on the same orbit (100 classes). For SGM, we keep the inference network identical to ours. Both use $SO(3)$ as hypothesis group. Error bars over multiple seeds will be included.
> > W2: ``Limited analysis of robustness and practical impact’’
>
> **Downstream tasks.** Important future work. Equivariant network literature already establishes that the correct group improves efficiency; our contribution is automating *which* group to use.
>
> **Sensitivity to prior.** $W_1$ distances are not comparable across hypothesis groups due to differing geometries. We introduce a **percent improvement over random sampling** metric ($1 - W_1 / W_{\text{random}}$) for scale-independent evaluation. Below we report $W_1$ (percent improvement) for all settings. Negative percent indicates worse than random.
>
> C4 arrow:
> | |LieGAN(1.gen)|LieGAN(4.gen)|LieFlow|
> |--|--|--|--|
> |SO(2) to C4|1.688(-211%)|1.672(-209%)|**0.072(86%)**|
> |GL(2) to C4(L)|1.688(81%)|1.672(81%)|**1.321(85%)**|
> |GL(2) to C4(M)|1.688(-90%)|1.672(-90%)|**0.428(51%)**|
>
> Irregular Tetrahedron:
> | |LieGAN|LieFlow|
> |--|--|--|
> |SO(3) to Tet|2.326(-90%)|**0.0885(92%)**|
> |SO(3) to Oct|2.198(-124%)|**0.0962(90%)**|
> |SO(3) to Ico|1.861(-154%)|**0.118(83%)**|
> |SO(3) to SO(2)(a)|0.0693(96%)|**0.0429(97%)**|
> |SO(3) to SO(2)(b)|0.0309(98%)|**0.0299(98%)**|
>
> ModelNet10 (with new baselines):
> | |LieGAN|Augerino|SGM|LieFlow|
> |--|--|--|--|--|
> |SO(3) to Tet|2.291(-87%)|1.173(4.1%)|1.132(7.3%)|**0.113(91%)**|
> |SO(3) to Oct|2.333(-127%)|1.032(-5.1%)|1.018(-3.1%)|**0.125(87%)**|
> |SO(3) to Ico|1.828(-150%)|1.081(-47%)|1.386(-89%)|**0.156(78%)**|
> |SO(3) to SO(2)(a)|**0.0291(98%)**|1.452(23%)|2.214(-16%)|0.0461(97%)|
> |SO(3) to SO(2)(b)|0.0621(97%)|1.455(23%)|2.372(-24%)|**0.0472(97%)**|
>
> LieFlow achieves substantial improvements across all settings, significantly outperforming all baselines on discrete groups.
>
> **Robustness.** Controlled noise experiments on ModelNet10 (Ico) show graceful degradation under both transformation noise and masking; see Reviewer 2EKR Q3 for the full table. MI-Motion (Reviewer 2EKR W1) further shows robustness on noisy real-world data.
> > W3: ``Minimal evaluation of partial or approximate symmetries’’
>
> **Partial symmetries.** The partial $C_4$ experiment is indeed input-dependent: the dataset contains only three orientations ($0, \pi/2, 3\pi/2$), so the valid subset $H_x$ differs by orientation. Alg. 3 samples from the conditional $q_\theta(g \mid x)$, which is input-dependent by construction. We will add per-orientation visualizations at Fig 2.1 in link <https://anonymous.4open.science/r/Lieflow_rebuttals-5A7D/rebuttals.md>.
>
> **Approximate symmetries.** Our masking experiments (Table 3) already break exact invariance. We have additionally added controlled noise experiments on ModelNet10 (Ico); please see our response to Q3 of Reviewer 2EKR for the full table. LieFlow degrades gracefully under both transformation noise (~3°–6° perturbations) and point masking (5–10%), demonstrating robustness to approximate symmetries. We will explicitly frame these as approximate symmetry evaluations in the revision.
>
> ## Questions
> > Q1: ``Sensitivity to the choice of hypothesis group’’
>
> If the hypothesis group $G$ does not contain the true symmetry group $H$, LieFlow can only recover the intersection $H \cap G$: it will learn whatever valid symmetries exist within $G$ but cannot discover transformations outside it. This limitation is shared by any method operating within a prescribed group, so we recommend erring on the side of a broader $G$. Our 2D experiments demonstrate this tradeoff: even with the much larger $GL(2, \mathbb{R})^+$, LieFlow still recovers $C_4$ and outperforms LieGAN (Table 1), though a tighter group like $SO(2)$ yields better absolute performance due to the stronger inductive bias. We will add guidance on hypothesis group selection in the revision.
>
> > Q2: ``Clarification of the Wasserstein-1 distance metric’’
>
> For two rotation matrices $R_1, R_2 \in SO(3)$, we use the geodesic distance $d_G(R_1, R_2) = \|\log(R_1^T R_2)\|_F / \sqrt{2}$, corresponding to the rotation angle between them. For $SO(2)$, this reduces to the absolute angular difference. $W_1$ is computed over these pairwise distances using the POT library. We chose $W_1$ because it captures both coverage and precision of the generated distribution.
> See W2 above for our new percent improvement metric that addresses cross-group comparability.
> ## Limitations
>
> We will expand the impact statement to discuss risks of incorrect symmetry identification in downstream models and recommend validating discovered symmetries against domain knowledge.

---

> > ### Author Rebuttal · Reviewer_5LQj · 2026-04-04
> >
> > The authors have provided thorough and detailed responses to my concerns, including new experiments and clarifications that address the points I raised. I will keep my scores but rise the confidence.

---

> > > ### Author Response · Authors · 2026-04-06
> > >
> > > We are glad the new experiments and clarifications were helpful. We thank the reviewer for their careful consideration and will make sure all improvements are incorporated in the final version.

---

### Official Review · Reviewer_2EKR · 2026-03-12

**Soundness:** 3
**Presentation:** 3
**Significance:** 2
**Originality:** 4
**Overall Recommendation:** 5
**Confidence:** 2

**Summary:**

This paper proposes a novel way to identify symmetries in data, by using the Flow Matching framework. The key idea is, rather than estimating the vector field that brings noise to the data (as done in standard flow matching), the authors propose to estimate the generator of the symmetry in the Lie Algebra of some larger (hypothesis) Lie group. The authors achieve this by proposing a training strategy which samples from the hypothesis group, creates an augmented sample by transforming a data sampling according to the prescribed transformation (associated with the group element) and then regressing the underlying Lie Algebra element which "explains" this transformations. The major difference between this setup and standard Flow Matching is that the authors are trying to recover the most likely transformations, which still preserve the underlying data distribution.

**Compliance With Llm Reviewing Policy:**

Affirmed.

**Final Justification:**

I thank the authors for their comprehensive answer and especially for including experiments on real-world data. My concerns have been largely resolved and I am happy to raise my score to Accept to reflect this. I encourage the authors to both include the additional results and also to improve the quality of the writing (addressing questions of presentation as highlighted by other reviewers as well).

**Key Questions For Authors:**

My main questions are related to the limitations:

1. Are there scenarios in which the method can infer the underlying symmetry group, which was not imposed artificially but rather encoded in the data itself?

2. In Section 4.1., the authors state "Although transformations are sampled from a broad hypothesis group during training, the flow-matching objective implicitly filters out transformations that are not true symmetries of the data. [...]" This seems to be an important claim, which serves as a basis for the entire paper. Can the authors provide a justification for this claim? I don't understand how "the flow-matching objective implicitly filters out transformations", in particular because in the original flow matching framework, the training happens with respect to "ground truth" flow (obtained via $\dot{x}_t = x_1 - x_0$), whereas in this paper's setting, the authors are sampling the transformations randomly from some hypothesis group. So, even if there is some filtering involved (e.g., because incorrect hypotheses cancel each other out), this is probably a property of the training dynamics rather than the flow matching objective itself. Some clarification would be very useful.

3. How robust is the underlying method to the noise _in the transformation group_? I see that the authors provided results in Table 3 on the Incomplete Observations. Furthermore, the authors provide experiments on "Partially Observed Symmetries" in Section 5.3. However, I wonder how the method behaves when there *is noise* in the provided transformations and moreover, how does it *degrade* in the presence of such noise and incomplete information. The results in Figure 2d do support the claim that the method is robust to some extent, but I think it would be very useful to understand the extent (and limits) of this robustness.

4. Can the authors discuss the computational complexity as a function of the hypothesis group? I wonder if the hypothesis group grows, the method might become intractable, since sampling in the vast majority of the hypothesis space would not lead to plausible transformations, and moreover the "cancellation effect" mentioned by the authors might not be visible until only a very large fraction of the hypothesis space is sampled.

**Limitations:**

I think this method presents some very interesting and original ideas, but its main limitations might be practical applicability, which is not entirely obvious from the submission itself.

The authors do provide a discussion of the limitations in the Conclusions section (Section 6) of the paper, and suggest some follow-up directions.

**Strengths And Weaknesses:**

From my perspective, the main strength of this paper is the highly original idea behind using flow matching for symmetry detection. Formulating the problem as a likelihood estimation *in transformation space* is both novel and can potentially lead to follow-up future work. The formulation via a hypothesis space and a Lie Algebra (which has vector space structure) on an otherwise complex Lie Group is also compelling and well-founded. The detailed analysis as well as a diverse set of experiments are also strengths of this paper.

The main weaknesses, as far as I can tell, are the fact that in *all* experiments, the results are shown on synthetic data, where the group structure is chosen a priori, rather than inferred directly from the data. The authors evaluate their approach on either synthetic 2D and 3D point clouds or, on ModelNet10. However, even in the latter scenario, the ground truth transformation group is actually selected by the authors, as described in Section 5.1, where the authors say "We consider five different target symmetry groups Tet, Oct, Ico, the SO(2) subgroup stabilizing the z axis ..." In other words, the authors control the target symmetry group and do not actually learn it from some real data. This makes me wonder whether there are actual practical scenarios where this method could be useful, and capable of inferring some hidden target symmetry group, which was not imposed by the authors. I have more related questions below in the Questions for the authors section. The authors state that their results "demonstrating the effectiveness in discovering symmetries in real-world datasets." However, I'm not 100% certain this is the case, since the target symmetry group is still specific by the authors.

---

> ### Author Rebuttal · Authors · 2026-03-31
>
> We thank the reviewer for their thoughtful feedback, particularly the focus on practical applicability.
>
> ## Weaknesses
>
> > W1: "The main weakness … all experiments are shown on synthetic data."
>
> We fully agree that demonstrating symmetry discovery without imposing the group is critical for validating practical applicability.
>
> To address this, we added a new experiment on the real-world MI-Motion dataset (Peng et al., 2023), which contains sequences of 3D human joint point clouds of multiple people interacting across different scenes (indoor, park, complex crowd, etc.). Crucially, we do not impose the target symmetry group; the symmetry structure must be inferred from the data itself.
>
> In this dataset, human trajectories are approximately canonicalized along the $x$- or $y$-axis in the world frame (people primarily move along axis-aligned directions), while gravity breaks full $SO(3)$ symmetry. This induces an approximate $C_4$ rotational symmetry around the $z$-axis, with occasional deviations due to interactions.
>
> Using $SO(3)$ as the hypothesis group, LieFlow correctly discovers that the data exhibits $C_4$ symmetry around the $z$-axis (see figure results at this link <https://anonymous.4open.science/r/Lieflow_rebuttals-5A7D/rebuttals.md>). The $SO(3)$ distribution visualization shows that the learned transformation samples concentrate on rotations around the $z$-axis. Extracting the $z$-axis rotation component and plotting the angle histogram clearly reveals four modes corresponding to the $C_4$ group elements, though with less peakiness than the synthetic $C_4$ arrow experiments, as expected for real-world data.
>
> In contrast, LieGAN only learns a uniform distribution over $z$-axis rotations, failing to discover the hidden discrete structure entirely.
>
> We will include these results and revise the wording to better distinguish settings where the group is specified vs. inferred.
>
> ## Questions
>
> > Q1: "Are there scenarios in which the method can infer the underlying symmetry group, which was not imposed artificially but rather encoded in the data itself?"
>
> Please see our response to W1 above, where we describe the new MI-Motion experiment that addresses exactly this scenario.
>
> > Q2: "flow-matching objective implicitly filters out transformations that are not true symmetries of the data…provide a justification for this claim"
>
> Please see our response to Reviewer s4QM W1, where we provide a formal justification. In short, our sampling procedure (Alg. 2) induces a generated distribution $\hat{p_\theta}$ over transformed data. Under standard realizability assumptions, the flow matching objective drives $\hat{p_\theta} \to p$, so at convergence only distribution-preserving transformations (i.e., true symmetries) remain in the support of the learned distribution, producing globally consistent training signals. We will formalize this into a proposition in the revision.
>
> > Q3: "How robust is the method to noise in the transformation group?"
>
> Great question. Our main paper already contains results on robustness to incomplete observations (Table 3) at 5% and 10% point masking. To further probe the limits, we added controlled noise experiments on ModelNet10 (Ico), applying random $SO(3)$ perturbations of varying magnitude after the true symmetry transformation:
>
> | Noise $\sigma$ | 0 | 0.05 (~3°) | 0.1 (~6°) |
> |---|---|---|---|
> | 0% mask | 0.156 | 0.272 | 0.425 |
> | 5% mask | 0.186 | 0.295 | 0.446 |
> | 10% mask | 0.212 | 0.326 | 0.474 |
>
> LieFlow degrades gracefully under both noise and masking, with performance remaining reasonable even at ~6° perturbation combined with 10% masking. The MI-Motion experiment (see W1) further demonstrates robustness on naturally noisy real-world data. We agree that characterizing precise failure boundaries is an important direction and will discuss this in the revision.
>
> > Q4: "Computational complexity as a function of the hypothesis group"
>
> This is a valid concern. Choosing a hypothesis group closer to the expected ground truth likely leads to faster convergence, as a tighter prior provides a stronger inductive bias. If one has prior information about the ground truth group, one should use it. Our 2D experiments support this intuition: within each hypothesis group setting, LieFlow significantly outperforms LieGAN (Table 1), but performance is better with the smaller $SO(2)$ than with $GL(2, \mathbb{R})^+$. In 3D, LieFlow successfully recovers discrete subgroups (Tet, Oct, Ico) from the full $SO(3)$ hypothesis group, suggesting the method scales reasonably even when most of the hypothesis space does not contain true symmetries.
>
> That said, scaling to very large or high-dimensional hypothesis groups remains an open challenge, and we will discuss this limitation more explicitly in the revision.
>
> ## Limitations
>
> We thank the reviewer for pointing this out. We will include a more complete discussion of limitations in the revision.

---

> > ### Author Rebuttal · Reviewer_2EKR · 2026-04-01
> >
> > I thank the authors for their comprehensive answer and especially for including experiments on real-world data. My concerns have been largely resolved and I am happy to raise my score to reflect this. I encourage the authors to both include the additional results and also to improve the quality of the writing (addressing questions of presentation as highlighted by other reviewers as well).

---

> > > ### Author Response · Authors · 2026-04-06
> > >
> > > We appreciate the reviewer's thorough engagement with our rebuttal and are glad to hear your concerns have been largely resolved. We will ensure the final version includes all new experiments and addresses the presentation feedback.

---

### Official Review · Reviewer_s4QM · 2026-03-13

**Soundness:** 3
**Presentation:** 3
**Significance:** 3
**Originality:** 3
**Overall Recommendation:** 4
**Confidence:** 3

**Summary:**

The paper employs flow matching for the task of discovering symmetries. The main idea is to perform flow matching over Lie groups instead of the data space. In this framework the learned flow induces a distribution over group elements whose support concentrates on the true unknown symmetry subgroup $H \leq G$ .

In the conditional flow matching objective, the velocity field corresponds to a Lie algebra generator $A$. Group elements are obtained through exponentiation of the generator, and orbit elements are generated by applying these group elements to the data.

The method is evaluated on datasets such as ModelNet10 and synthetic datasets. The proposed framework aims to handle both continuous and discrete symmetries.

**Compliance With Llm Reviewing Policy:**

Affirmed.

**Final Justification:**

Authors addressed my concerns regarding rigorous justification of symmetry discovery. Hence, I have raised my score to weak accept.

**Key Questions For Authors:**

Q-1) Why flow matching over Lie groups result in distribution concentrating on the symmetry group $H$ ? Could you please provide detailed explanation or derivation.

Q-2) Why the experiments were limited to unlabelled datasets? Is there such intrinsic limitation in the proposed method?

Q-3) How would the proposed method perform under noisy settings?

Q-4) Is there a way to extract the algebraic structure of the symmetry from the learnt distribution?

**Limitations:**

Yes

**Strengths And Weaknesses:**

# Strengths

**Originality**

The paper employs **flow matching in a novel and interesting way over Lie groups** (via a push-back vector field) for symmetry discovery. Instead of learning transformations directly in the data space, the method learns a distribution over group elements.

**Significance**

The ability to estimate **$H_x$** (in the case of partially observed symmetries) strengthens the significance of the work. In practical datasets we typically observe samples $x_1,\ldots,x_N \sim p(x)$, where each individual sample may only exhibit partial symmetry. Estimating $H_x$ for multiple samples could potentially be used to **recover the full symmetry group $H$**.

**Scope of applicability**

The framework is designed to handle **both continuous and discrete symmetries**, which broadens the scope of the method compared to approaches that focus only on continuous symmetries.

**Empirical results**

Although the experimental evaluation is somewhat limited, the reported results consistently show **superior performance of the proposed method** compared to the considered baselines.

---

# Weaknesses

**Soundness**

The paper provides an intuitive explanation that symmetry transformations produce **consistent training signals**, while non-symmetry transformations cancel out. However, the paper does not clearly explain **why the flow matching objective leads to recovery of the true symmetry subgroup $H$**. A clearer explanation or derivation of this mechanism would strengthen the technical soundness.

The approach relies on **exponential and logarithm maps** to move between Lie groups and Lie algebras. The paper does not discuss potential issues arising from these operations, such as branch ambiguities in the matrix logarithm or numerical stability considerations.

**Significance**

The experiments focus on **discovering symmetries from unlabeled geometric data distributions**. The paper does not consider settings where symmetry must be inferred from **labeled data through an unknown invariant or equivariant function $f$**. Studying such tasks would better demonstrate the applicability of the method to common machine learning problems involving symmetry.

**Presentation**

Lack of proper explaination on why the method lead to symmetry discovery. The existing explaination is what we already see in the works like LieGAN and Augerino.

While the authors briefly mention Augerino by Benton et al. (2020) in the introduction and related work, the paper would benefit from a much deeper empirical comparison against Augerino, particularly regarding how both methods handle partially observed symmetries.

The explanation in L172-187 (on the right column) that non-symmetry transformations produce “inconsistent directions that cancel out” is intuitive but not rigorous. The paper does not analyze the expected flow-matching objective to show that the symmetry subgroup corresponds to stationary points or optima. As written, the argument remains heuristic and does not establish that the learned distribution must concentrate on the true symmetry group.

# Comments / Suggestions

The evaluation metric is based on **Wasserstein-1 distance between sampled group elements**. While this is reasonable for comparing distributions over transformations, it evaluates sampled elements rather than intrinsic structural properties of the discovered symmetry.

For experiments involving **continuous symmetries** (e.g., $SO(3) \rightarrow SO(2)$), it may be useful to extract the learned **Lie subalgebra** and compare it with the true Lie subalgebra using metrics such as

- cosine similarity (for 1 dimensional Lie subalgebra)
- Grassmann distance
- projection distance
- principal angles between generator subspaces.

This would provide a more direct evaluation of whether the correct continuous symmetry direction is recovered.

---

> ### Author Rebuttal · Authors · 2026-03-31
>
> We thank the reviewer for their careful and constructive feedback.
> ## Weaknesses
> > W1: ``why the flow matching objective leads to recovery of the true symmetry subgroup.’’
>
> We agree a more rigorous justification is needed. Let $p$ be the data density, $G$ the hypothesis group, and $q_\theta(g \mid x)$ the learned conditional distribution. Alg. 2 produces $x’ = g \cdot x$ with $x \sim p$, $g \sim q_\theta(g \mid x)$, inducing the marginal $$\hat p_\theta(x') = \int_G \int_{\mathcal X} \delta(x' - g \cdot x) q_\theta(g \mid x) p(x) d\mu(g) dx.$$ Under standard realizability assumptions, flow matching objective drives $\hat p_\theta \to p$ (Lipman et al., 2023). At convergence, generated samples $x’ = gx \sim p$, implying that transformations in the support of $q_\theta$ preserve $p$, i.e. $g$ is a symmetry by definition. Thus, only transformations that induce distribution-preserving (consistent) training signals satisfy the objective globally while non-symmetries produce inconsistent targets that cannot match $p$. Moreover, exact recovery holds under a natural identifiability condition: if the only transformations in $G$ that preserve $p$ form a subgroup $H$, then $\textrm{supp}(q_\theta) \subseteq H$.
>
> Unlike LieGAN, our method needs no regularizer, as flow matching collapses from the prior to the correct symmetry, empirically finding the largest subgroup preserving $p$. We will formalize this in revision.
> > W2: ``relies on exponential and logarithm maps’’
>
> For the connected compact groups we consider ($SO(2), SO(3)$), the exponential map is well-defined. The logarithm map has branch ambiguities (e.g., near $\pm \pi$ in $SO(3)$), but these did not cause training instability in practice. Specialized implementations (e.g., Rodrigues formula for $SO(3)$) could improve numerical robustness. We will add this discussion.
> > W3: ``does not consider settings … labeled data’’
>
> We focused on the unsupervised setting as it is more general. However, LieFlow naturally extends to labeled settings by learning symmetries of the joint distribution $p(x, y)$. We ran a new experiment on the $C_4$ arrow dataset where {0°, 90°} are labeled 0 and {180°, 270°} are labeled 1. We concatenate the one-hot label to point positions to form a 4D point cloud and use $SO(2)\oplus SO(2)$ as the hypothesis group, where the first factor acts on position and the second on the label dimension. As shown in Fig2.2 at <https://anonymous.4open.science/r/Lieflow_rebuttals-5A7D/rebuttals.md>, the learned conditional distribution correctly distinguishes label-preserving rotations from label-changing ones for each sample, recovering the functional partial symmetry. We will include this experiment and discussion in the revision.
> > W4: ``The existing explaination is ... like LieGAN and Augerino’’
>
> LieGAN relies on a discriminator to distinguish real from generated data, providing no direct link between the objective and symmetry recovery. Augerino uses task loss to learn augmentation distributions, which is supervised and does not discover symmetries from unlabeled data.
> In contrast, LieFlow's flow matching objective directly enforces that the induced marginal matches the data distribution, yielding the identifiability argument in W1. We will clarify this distinction.
> > W5: ``empirical comparison against Augerino’’
>
> We have implemented both Augerino and SGM baselines and LieFlow significantly outperforms both across all settings (see Reviewer 5LQj W2 for the full results)
> >W6: ``.... intuitive but not rigorous’’
>
> See W1
>
> ## Comments
> > C1: "Wasserstein-1 distance…evaluates sampled elements"
>
> We agree. We have added a percent improvement metric (1 - $W_1/W_{\text{random}} $). See Reviewer 5LQj W2 for full tables.
> > C2: ``extract the learned Lie subalgebra…’’
>
> This is a great point. For the $SO(2)\subset SO(3)$ case of rotations about the z-axis, we sample transformations with Alg. 3, map them to $\mathfrak{so}(3)$, and use PCA to extract the learned 1D subalgebra. Its agreement with the ground truth is very strong: cosine similarity 0.9997, principal angle $1.256^\circ$, Grassmann distance $0.02193$ rad, and projection distance $0.03102$. This confirms that LieFlow recovers the correct continuous symmetry direction.
>
> ## Questions
> > Q1-Q2
>
> See W1 and W3.
> > Q3: ``noisy settings’’
>
>
> Table 3 shows robustness to 5% and 10% masking. We have added controlled noise experiments on ModelNet10 (Ico); LieFlow degrades gracefully. See Reviewer 2EKR Q3 for the full table."
>
> > Q4: ``extract the algebraic structure of the symmetry’’
>
> This is an interesting direction for future work. By thresholding the learned distribution over $G$, we can approximate its support and extract a finite set of group elements. Recovering the subgroup’s algebraic structure or isomorphism class is nontrivial in general, but computational group theory tools such as GAP can help identify generators, relations, and candidate groups, especially when the set of possible symmetries is restricted.

---

> > ### Author Rebuttal · Reviewer_s4QM · 2026-04-02
> >
> > Thank you for the rebuttal. I am still trying to understand the rigorous justification provided. At this stage, I will maintain my score. I will get back in a day or two.
> >
> > Update: My concerns are addressed. I now understood the derivation. I will raise my score to weak accept.

---

> > > ### Author Response · Authors · 2026-04-06
> > >
> > > We thank the reviewer for taking the time to carefully re-evaluate our rebuttal and for raising their score. We will incorporate all revisions in the final version.

---

### Official Review · Reviewer_CbhW · 2026-03-23

**Soundness:** 2
**Presentation:** 3
**Significance:** 3
**Originality:** 3
**Overall Recommendation:** 4
**Confidence:** 3

**Summary:**

This paper studies symmetry discovery from data and proposes LieFlow, a method that reframes the task as learning a distribution over a hypothesis Lie group $G$, instead of directly estimating symmetry generators. The idea is that the support of a learned distribution would concentrate on a true underlying symmetry subgroup $H\subset G$. To learn the flow directly in the group space, the method first samples a transformation $g$ from a prior over $G$, then applies it to a data point $x_1$ to form $x_0=gx_1$, constructing an interpolation along the group orbit via $x_t=\exp(tA)x_0$ where $A=\log(g^{-1})$, then trains a conditional flow-matching model to predict the corresponding Lie-algebra element.

A second technical contribution in this paper is that the authors identified a a “last-minute convergence” issue for discrete symmetry discovery. The authors observed that because flow vectors associated with different modes can cancel, the learned dynamics may stay nearly stationary until late in the trajectory. The paper then proposes a power time schedule with biases training toward later times to address this.

For the experiments, the paper evaluates LieFlow on synthetic 2D and 3D point-cloud datasets and on rotated ModelNet10, comparing primarily against LieGAN. The reported results indicate strong performance on discrete subgroup discovery, while continuous-group results are also included.

**Compliance With Llm Reviewing Policy:**

Affirmed.

**Key Questions For Authors:**

- *What exactly is identified by the objective?*
The paper’s central claim is that the support of the learned distribution over the hypothesis group $G$ recovers the true subgroup
$H$. However, the appendix repeatedly describes the generated trajectories as converging to the closest group/orbit element, which suggests a potentially weaker explanation of the results. Could the authors clarify why the current evidence should be interpreted as subgroup recovery rather than nearest-valid-orbit projection, and whether they can state explicit assumptions under which support recovery is identifiable?
- *How essential is the trivial-stabilizer assumption in the derivation?* In Appendix A, the identification of the target tangent vector with a unique Lie-algebra coordinate $A$ appears to rely on $J_{x_t}$ being bijective, which in turn holds when the stabilizer is trivial. Since many symmetric objects of practical interest have nontrivial stabilizers, could the authors explain whether the method and its interpretation still go through in that case, and what the learning target should be when the Lie-algebra coordinate is not unique?
- *Can the authors reconcile Section 4.3 with the partial $C_4$* experiment?
Section 4.3 formalizes partial symmetry as an input-dependent subset $H_x\subset H$ and interprets the method as learning a conditional distribution $p_\theta(\cdot|x)$ whose support reveals $H_x$. In contrast, the partial $C_4$* experiment appears to use the same fixed subset of rotations for every sample and then treats recovery of the full closure as success. Could the authors clarify whether the method is actually demonstrated on sample-dependent partial symmetries, or whether the current evidence should be interpreted more narrowly as closure completion under missing transformations?
- *Where, concretely, is the evidence for approximate symmetry?*
The paper explicitly claims coverage of approximate symmetries, but I mainly saw experiments on exact subgroup recovery, a partial $C_4$ setting, and robustness to masking/noisy observations. Which experiment should the reader regard as evidence for approximate-symmetry discovery in the sense claimed in the contribution list? If none directly supports that claim, would the authors be willing to narrow the statement accordingly?
- *How robust is the method to the choice of ambient group and prior?*
- *Why is the empirical comparison limited to LieGAN, and how stable are the reported numbers?*

**Limitations:**

No. The paper does acknowledge some real limitations, such as small point clouds, a limited number of objects, scalability to broader modalities and higher-dimensional/non-compact groups, and sensitivity to the prior-group parameterization, but the discussion is still incomplete. It would be stronger if the authors explicitly discussed the dependence on the chosen hypothesis group and prior, the controlled nature of the ModelNet10 setup (canonical objects with sampled known transformations), and the gap between the broad claims around approximate / partially observed symmetries and the current evidence.

**Strengths And Weaknesses:**

Strength:
- The paper has a genuinely interesting core idea. Recasting symmetry discovery as support learning in group space, rather than direct generator estimation is conceptually fresh and the flow-matching formulation on a Lie group is noval.
- The appendix analysis of “last-minute mode convergence” is also one of the stronger parts of the paper. It gives a plausible explanation for why discrete subgroup recovery is hard and why skewing time samples toward $t\approx1$ helps.
- The problem studied in this paper is important, and a practical method for discovering discrete symmetries would be interesting to a broad range of ICML audience. The power-schedule ablation developed in this paper could be useful as well.

Weakness:
- The paper’s central claim is that 'non-symmetry transformations induce inconsistent targets that cancel, while true symmetries induce coherent targets, so the learned distribution concentrates on the true subgroup'. This is intuitively appealing, but it is not turned into a theorem, an identifiability statement, or even a precise proposition with explicit assumptions. The main scientific claim is heuristic. In Appendix A, there is a hidden technical restriction that the identification of the target tangent vector with a Lie-algebra coordinate $A$ is justified through a bijection which the appendix explicitly ties to a trivial stabilizer assumption. This is not a minor technicality but a real restriction on when the target is well defined, and this was not foregrounded in the main paper.
- A second soundness concern is that the appendix’s own analysis points to a weaker explanation of the empirical results than the paper suggests. In Appendix D and the later qualitative appendices, the learned trajectories are repeatedly described as converging to the closest orbit element, not necessarily recovering the original $x_1$. The posterior remains near-uniform until very late times and then collapses to the nearest mode. That behavior is fully consistent with the observed low Wasserstein distances on these orbit-generated benchmarks, but it leaves open whether the method is really recovering subgroup structure in a strong sense, or whether it is learning a nearest-valid-orbit projection rule that works well on highly structured synthetic tasks.
-  The significance is somewhat overstated relative to the current experimental setting. Even the “real-world” benchmark is still generated in a controlled way: the paper selects 10 canonical ModelNet10 objects per class, downsamples them to 128 points, and then creates the dataset by applying sampled target-group transformations. This is a reasonable benchmark, but it is still a synthetic orbit-generation problem on curated 3D shapes, not discovery of latent symmetries in naturally occurring data. From this perspective, the paper demonstrates promise in a well-specified regime, rather than a broadly validated solution to symmetry discovery in the wild.
- The same issue appears in the claims about partial and approximate symmetry. Section 4.3 formalizes partial symmetry using input-dependent subsets $H_x\subset H$, but the empirical 'partial $C_4$' experiment uses the same three transformations for every sample and then treats recovery of the full closure $C_4$ as success. That is not really a demonstration of identifying sample-dependent $H_x$. Likewise, although the introduction and contribution list claim coverage of approximate symmetries, I did not see a dedicated approximate-symmetry benchmark. So the empirical evidence supports exact subgroup recovery under controlled orbit generation much more strongly than it supports the broader headline.
- For the experiments, the paper compares only against LieGAN, despite discussing several nearby papers in the related-work section. The tables report single numbers without seed variability or uncertainty bars. This matters especially because the method appears quite sensitive to the choice of ambient group and prior parameterization: for $SO(2)\rightarrow C_4$, the reported Wasserstein distance is
0.072 while for $GL(2,\mathbb{R}^+)\rightarrow C_4$ it is 1.321 under one prior and 0.428 under another. This is a very large spread, and it suggests that prior design and hypothesis group choice are doing substantial work.

---

> ### Author Rebuttal · Authors · 2026-03-31
>
> We thank the reviewer for their thoughtful and insightful review.
>
> ## Weaknesses
>
> > W1: "The paper's central claim…heuristic"
>
> Great point. We agree to provide a formalization. Let $p$ denote the data density, $G$ the hypothesis group, and $q_\theta(g \mid x)$ the learned conditional. Alg. 2 produces samples $x' = g \cdot x$ with $x \sim p$, $g \sim q_\theta(g \mid x)$, inducing marginal $$\hat p_\theta(x') = \int_G \int_{\mathcal X} \delta(x' - g \cdot x) q_\theta(g \mid x) p(x) d\mu(g) dx.$$ Under standard realizability conditions, the flow matching objective drives $\hat p_\theta \to p$ (Lipman et al., 2023). At convergence, $x' = gx \sim p$, so transformations in $\text{supp}(q_\theta)$ must preserve $p$. If the only $p$-preserving transformations in $G$ form $H$, then $\text{supp}(q_\theta) \subseteq H$. We will formalize this into a proposition. See also Reviewer s4QM W1.
> > W2: "trivial stabilizer assumption"
>
> We thank the reviewer for this insightful observation. The trivial stabilizer assumption as stated in Appendix A is stronger than necessary. Upon closer analysis, what is actually required is the weaker condition that the *infinitesimal* stabilizer be trivial, i.e., $\text{Lie}(G_x) = \{0\}$, ensuring the map $A \mapsto J_x(A)$ is injective. This is a strictly weaker assumption that significantly broadens applicability: it holds whenever the stabilizer is discrete, covering all finite subgroups. We confirm this on a $C_4$ rectangle with $C_2$ as self-symmetry, where LieFlow correctly recovers the full $C_4$ (Fig3.1 at <https://anonymous.4open.science/r/Lieflow_rebuttals-5A7D/rebuttals.md>), precisely because $\text{Lie}(C_2) = \{0\}$ introduces no degeneracy at the Lie algebra level. The method encounters theoretical limitations only when the stabilizer contains a *continuous* subgroup which introduces a non-trivial kernel. For the strongest guarantees in such settings, auxiliary information such as labels would be needed. We will revise Appendix A with the corrected assumption and include the rectangle experiment in the revision.
> > W3: "weaker explanation…nearest-valid-orbit-projection rule"
>
> While it is true that individual trajectories often exhibit ``nearest orbit’’ projection, we disagree that this means our method only works on highly structured tasks or fails to discover subgroup structure.  In Alg. 3, because the prior is broad over $G$, different initial samples $g \sim p(G)$ induce trajectories that converge to *different* valid orbit elements. For a fixed $x_1 \sim q$, varying the prior yields multiple distinct transformations $Mg = h \in H$, rather than a single projection. As $x_1$ varies as well, this allows the model to recover the full subgroup support, not just a nearest representative.
> > W4: "…symmetry discovery in the wild"
>
> We fully agree. We added an experiment on MI-Motion (Peng et al., 2023), a real-world motion-capture dataset. LieFlow discovers $C_4$ from $SO(3)$; LieGAN only recovers uniform $SO(2)$. See Reviewer 2EKR W1 for details.
> > W5: "partial symmetry…sample-dependent…approximate symmetries"
>
> **Partial:** We agree that the input-dependent distribution should be shown to demonstrate learning partial symmetry. The visualization of the sample-dependent distributions for three different arrow orientations is shown at Fig2.1 in <https://anonymous.4open.science/r/Lieflow_rebuttals-5A7D/rebuttals.md>, and LieFlow successfully discovered the support for each orientated arrow. In addition, we have added a new experiment on learning functional partial symmetry. See Reviewer s4QM W3 for more details. We will add these experiments in the revision.
>
> **Approximate:** We acknowledge this point and will narrow our claim. Our masking (Table 3) and noise experiments (see Reviewer s4QM Q3 for the full table) demonstrate *robustness under approximate symmetry* rather than discovery of approximate symmetry. We will revise the contribution list accordingly.
> > W6: "only against LieGAN…large spread"
>
> We have added Augerino and SGM baselines; LieFlow significantly outperforms both. Error bars will be included. Regarding the spread: $W_1$ distances are not comparable across hypothesis groups due to differing geometries; To align this divergence, we introduce a percent improvement over random sampling metric for scale-independent evaluation. See Reviewer 5LQj W2 for full results and detail.
> ## Questions
> > Q1–Q4
>
> Addressed in W1, W3, and W5.
> > Q5: "How robust ... of the ambient group and prior"
>
> A tighter hypothesis group improves performance due to stronger inductive bias (Section 5.3), but LieFlow is fairly robust: even with the larger $GL(2, \mathbb{R})^+$, it recovers $C_4$ and outperforms LieGAN (Table 1). Current evidence is limited to 2D; we will clarify.
> > Q6: ``empirical comparison limited to LieGAN’’
>
> See W6.
> ## Limitations
> We appreciate the reviewer for highlighting this, and we will add a fuller discussion of the limitations in the revision.

---

> > ### Author Rebuttal · Reviewer_CbhW · 2026-04-01
> >
> > Thank you for substantially addressing my concerns. I would revise my rating to a Weak Accept.

---

> > > ### Author Response · Authors · 2026-04-06
> > >
> > > We are happy to hear your concerns have been fully addressed and thank you for raising your score. We will ensure all revisions are incorporated in the final version.

---

### Decision · Program_Chairs · 2026-04-30

**Decision:**

Accept (regular)

**Comment:**

This submission obtained unanimously positive scores: one accept and three weak accepts. While the main concerns were unclear presentation and limited experimentation, the authors provided a good rebuttal with additional experiments, assuaging most of the concerns. All the reviewers acknowledged the novel flow-matching approach for symmetry discovery. Siding with the consensus, AC also recommends acceptance.